# Simultaneously Enhanced Strength and Ductility of Al–Mg–Si Alloys during Aging Process Induced by Electro-Pulsing Treatment

**DOI:** 10.3390/ma12091383

**Published:** 2019-04-28

**Authors:** Yitong Wang, Yuguang Zhao, Xiaofeng Xu, Dong Pan, Wenqiang Jiang, Xueying Chong

**Affiliations:** Key Laboratory of Automobile Materials, Ministry of Education and School of Materials Science and Engineering, Jilin University, No. 5988 Renmin Street, Changchun 130022, China; wyj17@mails.jlu.edu.cn (Y.W.); zhaoyg@jlu.edu.cn (Y.Z.); pandong16@mails.jlu.edu.cn (D.P.); jiangwq16@mails.jlu.edu.cn (W.J.); zhongxy17@mails.jlu.edu.cn (X.C.)

**Keywords:** 6061 alloy, electro-pulsing treatment, clusters, sub-grains, APT

## Abstract

Most methods used for strengthening metallic materials, such as thermal-mechanical treatment, will sacrifice the ductility. A novel technology, electric pulsing treatment (EPT), is applied to break this trade-off, which produces an Al–Mg–Si alloy with superior ductility and higher strength within only 560 ms. Systematic electron microscopy characterization and finite element simulation reveal that EPT promotes the formation of clusters Mg_2_(Si,Cu)_3_ and sub-grain boundaries. The results of quantitative calculation indicate that the dislocation entanglement is delayed due to the existence of clusters and longer dislocation glide distance, so that ultimate strength is fully improved. Moreover, the superior ductility is mainly governed by sub-grains which lead to higher mobile dislocation density, appearance of new crystal orientations, and prevention of crack propagation. Thereupon, this interesting finding paves the way in developing the Al–Mg–Si alloy with higher mechanical properties efficiently.

## 1. Introduction

Al–Mg–Si alloys are promising structural materials to develop automobile manufacturing and railway components because of the combination of lightweight and advanced anti-corrosion [1]. The major strengthening approach for this alloy is the Orowan mechanism, namely the strength increment is caused by interactions between dislocations and acicular monoclinic Mg_2_Si phase [2]. Generally, the strength and ductility of these alloys are mutually exclusive (trade-off relationship) during the aging process i.e., increment of strength usually accompany with decrease in ductility. Therefore, the comprehensive mechanical properties cannot be simultaneously improved to satisfy the requirements to further develop the technology of transportation industry.

The grain refinement strengthening was deemed to be a method which can break the trade-off relationship. Deformation combined with thermal treatment, as a widely adopted grain refinement approach through dynamic recrystallization, can enhance the mechanical properties effectively. Whereas, due to the higher stacking fault energy of α-Al matrix, the requirements for thermal processing parameter and material parameter (temperature and deformation amount) are more stringent [3]. Other advanced technologies, such as ECAE (equal channel angular extrusion), etc., can offer an effective procedure to obtain bulk nanometer scale materials by intensive deformation. However, in aluminum alloys, even though ultrafine-grained structures are derived, substantial crystal defects are also obtained. In this way, the ductility was relatively less desirable [4]. Besides, the lattice coherence between β″ phase and α-Al matrix will gradually decrease during long time traditional aging process, which will promote the crack initiation during tensile deformation. In other words, traditional aging treatment and the aforementioned refinement methods are not only less effective because of the complicated process, but also time-wasting for Al–Mg–Si alloys. Therefore, a new high-efficiency treating method is urgently needed.

Fortunately, electro-pulsing treatment (EPT) is a process in which the metallic materials are treated in the form of thermal–stress–electric coupling in an extremely short period, which can make obvious changes in the microstructure evolution and mechanical properties, such as self-healing of damage inside [5], relaxation of structure [6], refinement of microstructure [7], gradient distribution of mechanical properties [8], solidification of liquid structure [9], and so on. Especially, electro-pulsing can enhance the strength and maintained the plasticity of metallic materials simultaneously. In Zhao’s research [10], boron steel with relatively better plasticity was devised under electro-pulsing. Besides, An et al. [11] reported that the products of strength-ductility of GH4169 alloy were remarkably modified by electro-pulsing current due to the enhanced vacancy concentration. Moreover, Wang [12] argued that the ductility of TA15 titanium alloy increased 122% with a non-obvious decreased tensile strength after pulse current was applied. Hence, the above-mentioned researches imply that the simultaneously improved ductility and tensile strength can be achieved by high-energy electro-pulsing treatment.

In this report, high-energy electro-pulsing treatment is designed to get advanced combination of ductility and strength of 6061 alloy (a kind of Al–Mg–Si alloy). In contrast to artificial aging (T6), microstructural evolution and precipitation behaviors between different samples are fully characterized by optical microscopy (OM), electron scanning microscope (SEM), transmission electron microscopy (TEM), electron backscatter diffraction (EBSD), atom probe tomograghy (APT), and X-ray diffraction (XRD). Combined with finite element model (FEM) simulation and quantitative calculation, some especially interesting phenomena will also be discussed by deeply exploring the relationship between microstructure and properties. It is expected to figure out the progressiveness of electro-pulsing in the field of strengthening and toughening of metallic materials.

## 2. Materials and Methods

The composition of commercial AA6061 is displayed in Table 1. The original AA6061 was rolled and annealed, and the sheet thickness was 10 mm. The 6061 sheets were sliced into 65 mm lengths by a wire-cutting machine and the cross-section area of machined slices was 30 mm^2^ (3 × 10 mm^2^).

Solution (S) treatment was executed by Nitrogen oven and the treating temperature and procedure were 550 °C and 1 h, respectively. The electro-pulsing treatment was carried out by a self-made thyristor circuit with air cooling, and Figure 1a shows the detailed framework of this machine. The average voltage and current density in this paper are 3 V and 5.146 × 10^7^ A/m^2^. As to traditional aging process (T6), its parameters are 175 °C and 8 h. Whereas, for electro-pulsing treated samples, the treating procedure is 560 ms just provided by the electric pulsing machine (referred to as S + EA, S for solution, EA for electro-pulsing treatment followed with air cooling). Figure 1b represents the specific treating technology.

A tensile properties test was conducted on MTS-810 system at 25 °C (tensile rate, 0.3 mm/min), and there were 8 samples for tensile test in each condition. The elongation was measured by a BCX5000 extensometer (Shanghai LEAO Test Instrument Co., Ltd., Shanghai, China) matched to the tensile test equipment. The detailed size of samples for tensile test is shown in Figure 1c.

D/Max 2500PC XRD machine (Rigaku Corporation, Tokyo, Japan) was applied for analyzing the phase constitutions of different samples. The scanning range was 10°–90° and the speed was 2°/min. The ZEISS AxioImager A2m optical microscopy (Carl Zeiss AG, Oberkochen, Germany) was used to detect the grain size. The JEM-2100 microscopy (JEOL LTD, Tokyo, Japan) was adopted to detect the microstructural evolution and aging behaviors. The distribution of local misorientation was characterized by EBSD test on a ZEISS EVO 18 SEM microscopy (Carl Zeiss AG, Oberkochen, Germany). Rigaku D/max-rA XRD machine (Rigaku Corporation, Tokyo, Japan) was used to measure the dislocation density and Rietveld software was to quantitatively analyze the XRD patterns (scanning range 30°–85°, step 0.02°). APT detection was operated by a LEAP 4000 machine (CHAUVIN ARNOUX GROUP, Paris, France) and samples for this test were processed into ultra-fine rods (section size: 0.5 × 0.5 mm^2^, and length dimension is 20 mm) [13]. IVSA^TM^ software was used for the statistics of APT data. COMSOL Multiphysics® software was adopted for the simulation of electro-pulsing treatment.

## 3. Results and Discussion

### 3.1. Mechanical Properties and Work-Hardening Rate

Figure 2a shows the typical strength–strain engineering curves of samples in different states. The detailed data, including the strength–ductility balance [14], are shown in Table 2. The results indicate that after traditional aging treatment, the UTS (ultimate tensile strength) is greatly improved (from 196.5 ± 3.0 MPa in the S state to 286.1 ± 1.9 MPa in the T6 state), but the ductility is obviously decreased (from 38.3 ± 1.4% to 22.7 ± 1.2%). Whereas, after electro-pulsing treatment, the UTS of S + EA sample (292.6 ± 2.1 MPa) is a little higher than the T6 sample. Furthermore, the ductility (40.6 ± 0.9%) is even higher than S sample. It is interesting that although the UTS of S + EA sample is higher, its yield strength (YS) is obviously lower than T6 sample. This interesting phenomenon will be discussed in the later section.

The normalized work-hardening rate (*Θ*, (*dσ_true_*/*dε_true_*)/*σ_true_*) curves are plotted in Figure 2b. The *σ_true_* means true stress and *ε_true_* refers to true strain [15]. As is seen, compared with T6 sample, the S + EA sample possesses a larger work-hardening rate. It is worth noting that the T6 sample presents continuous stress relaxation whereas there is a small increase in work-hardening rate of S + EA sample before the fracture. In general, as is seen in Table 2, the strength–ductility balance of S + EA sample is 5385 MPa × % higher than T6 sample, which indicates that electro-pulsing treatment has more obvious advantages on the modification of 6061 alloys.

### 3.2. Microstructure and Precipitates Characterization

The aforementioned results indicate that mechanical properties of 6061 alloys can be remarkably enhanced by electro-pulsing treatment. To deeply explore the effect of pulse current on tensile properties of 6061 alloys, microstructural differences between different samples need to be systematically characterized.

XRD spectrums of different samples are depicted in Figure 3 to detect the phase composition of different samples. Characteristic peaks of Mg_2_Si and Al_5_FeSi can be distinguished in T6 samples, but in the S + EA samples, the characteristic peaks of these precipitates cannot be seen clearly, which may be related to the limited content of precipitates.

In addition, the peaks in XRD pattern of S + EA sample shift slight to the right of T6 sample. Besides, the corresponding FWHM (full width half maximum) of S + EA sample is also wider than T6 sample. Consequently, it is concluded that there will be little precipitates in S + EA sample, and finer grain combined with greater lattice distortion may also exist. Therefore, necessary EBSD observations need to be conducted.

EBSD observation is applied to deeply understand the microstructural evolution in different samples. Figure 4a–c depicts the optical images of different samples. As displayed in Figure 4d–f, different colors represent different crystal orientations, the bold black lines represent grain boundaries higher than 10° and the thin gray lines represent grain boundaries less than 2°. Figure 4g–i demonstrates the distribution of local misorientation (KAM (kernel average misorientation)). It is clearly seen that after T6 treatment, the number of low misorientation boundaries is remarkably decreased. This can be attributed to the recovery of dislocations in S sample during the long aging process. As for S + EA sample, the number of low misorientation boundaries increases significantly, as shown in Figure 4f. This indicates that after electro-pulsing treatment, there will be an increment in dislocation density in S + EA sample [16]. In this way, the lattice distortion will also become more severe. Thus, there appears a deviation toward right in XRD pattern of S + EA sample (Figure 3).

Different from T6 sample, finer grains with new crystal orientations appear in the interior of prior α-Al grains. This is consistent with the wider FWHM of S + EA sample in Figure 3. For the formation of these finer grains, it is related to the new generated dislocations. In fact, during the process of electro-pulsing treatment, the distribution of current density is heterogeneous due to the diverse microstructures of polycrystalline metallic materials. In high-current density areas, the dislocations are easier to tangle with each other to form a structure closer to sub-grain boundary under the effect of electro-migration [17]. Therefore, new orientations will appear. 

Precipitation behaviors of aging treated samples are the basic reasons for the difference of mechanical properties. Also, to further investigate the microstructural difference revealed by XRD and EBSD analysis, TEM observation is carried out.

As depicted in Figure 5a–c, granular Al_5_FeSi phases and needle-like Mg_2_Si phases are detected. The corresponding FFT (fast Fourier-transform) inserts in Figure 5b,c implies the accurate crystalline structure of the above-mentioned phases, which verify the results in Figure 3. Whereas, in S + EA sample, no precipitation is detected based on the SAED (selected area electronic diffraction) and only diffraction spots of matrix are observed. Besides, a large amount of sub-grains is also founded. Combining the results of XRD and EBSD, it can be concluded that the wider FWHM, the increment in frequency of local misorientation, and the new orientations in S + EA sample are all related with the formation of sub-grains. As for the cell-like structure of sub-grains, the corresponding results of HRTEM (high-resolution transmission electron microscopy) are given in Figure 5e. As is seen, the FFT insert of this area shows that there are mixed sets of diffraction spots of the cell wall (sub-grain boundary). This indicates that sub-grain boundaries are formed by the dislocation entanglement. Moreover, for the interior of sub-grain, some globular clusters are detected. FFT results indicate that the crystal structures of these clusters are almost the same as the matrix. As is reported in Reference [18], the general de-solution sequence is atomic clusters→GP zones→precipitations. Even though the structure of clusters is the same as the matrix, there still exists distinct difference of chemical component between them.

APT test is conducted to clarify the composition of clusters in S + EA sample. Figure 6a shows 3D reconstructions of different atomic positions. Figure 6b displays the distributions of the clusters, which indicates the clusters may consist of Mg, Si, and Cu.

Figure 7a–f presents the nearest neighbor distribution of Cu, Mg, Si, Al, Mn, and Cr, in which the red line represents the distribution curves of the distance between the two nearest neighbors if the atoms are randomly distributed, while the black line represents the distance distribution curves between the two nearest neighbors in the actual test data collection. If the black line deviates to the left of the red line, it indicates that the atomic spacing of the actual test data is less than the random distribution, and this element is enriched. If not (whether the black line coincides with red line or black line deviates to the right), it means that the actual test data are consistent with the random distribution, and the element is randomly distributed. From Figure 6 and Figure 7, the results demonstrate that there is an evidently decomposition appears after electro-pulsing is applied. From the figures mentioned above, the segregation of Mg, Si, and Cu is detected. Besides, as is counted by IVSA^TM^ 3.6.12 software, a total of 166 clusters are observed, the volume fraction is 7.26% and the mean size (considered as a sphere) is 1.3 nm. Detailed atomic numbesr of Mg, Si, and Cu in all clusters are listed in Table 3. Finally, the average composition of these clusters is determined to be Mg_2_(Si,Cu)_3_. Besides, for the T6 sample, from the results of HRTEM figure (Figure 5b), there only exist Mg_2_Si and Al_5_FeSi phases. In this way, the APT test of T6 sample is not considered.

### 3.3. The Formation Mechanism of Sub-Grains in S + EA Sample

The results of Figure 4 and Figure 5d indicate that there exist a lot of sub-grain boundaries in S + EA sample. To further research the effect of pulse current on formation of sub-grains, finite element modeling simulation was conducted. 

Before the simulation, some necessary models should be built. As is seen in Figure 8a, the α-Al matrix is simplified as a cube (1000 mm^3^) and dislocation line is regarded as a cylinder (Φ 5 mm, 2 mm length). Besides, based on classical dislocation theory, due to the fact that lattice distortion exists around the dislocations, the conductivity of dislocation line in this model is lower than the matrix. What is more, the angle between current direction and direction of dislocation line should also be ascertained. Therein, the simulation model can be classified into two types: (i) The current direction and the direction of dislocation line are parallel to each other (model 1); (ii) the angle between current direction and direction of dislocation line is 90° (perpendicular to each other, model 2). The meshing method for the model is standard meshing (Figure 8b).

For model 1, the simulation results are shown in Figure 8d–j. As is seen, there exist orbicular high-current density and temperature area in the plane perpendicular to current direction. What is more, the corresponding thermal stress distribution in the same area is relatively lower than other areas. Moreover, the highest thermal stress is located at both ends of the dislocation line.

As for model 2, the simulation results are displayed in Figure 9a–g. It is clearly found that the temperature and current density at YZ plane are obviously higher than other areas. Whereas, the thermal compressive stress in this area is relatively lower than other areas. Furthermore, the lowest thermal stress is observed at both ends of the dislocation line.

Considering the effects of the coupled three fields mentioned above, the detailed changes on the morphology of dislocations are given in schematics (Figure 9h,i). In model 1, in view of electro-migration effect and atomic thermal diffusion, the atoms in the annular area are easier to diffuse. Additionally, the both ends of dislocation line can be considered to be fixed. Therefore, after pulse current is applied, the dislocation line in model 1 can be regarded as Frank-Read dislocation source. In this way, plenty of dislocation cells will be formed continuously. On the other hand, in model 2, the atoms in the annular area are also easier to diffuse. But because the thermal stress at both ends of dislocation lines is the lowest (considered as unconstrained), the dislocations can only glide in YZ plane. The comprehensive results of dislocation motion in models 1 and 2 are intersection and annihilation of dislocations, respectively. In fact, the angle between current direction and direction of dislocation line is a random value at a range of 0°–90°. Therefore, to reduce the whole system energy, the actual dislocation motion tends to form a structure with an enclosed geometry shape, just like sub-grains.

From a qualitative perspective, the amount of sub-grains in S + EA sample is higher than T6 sample. However, the treating duration is only 560 ms, which is not sufficient enough to form so many sub-grains. So, it is necessary to analyze the formation mechanism quantitatively.

It is widely accepted that the dislocation movement in cold deformed metallic materials can be accelerated by EPT [19]. Under the effects of electron wind force induced by pulse current, by which the dislocations are scattered around unevenly, then, the barrier between adjacent dislocations are reduced [20]. By this, the enhanced mobility of dislocations is [21]:(1)$$J=\frac{2\;NeZ^{*}D\rho \;j_{m}f\;\tau_{p}}{K\;T}.$$

In Equation (1), *J* represents atomic diffusion flux, *N* refers to atomic density, *e* means the electric charge quantity of an electron, *Z*^*^ is quantivalence of Al, *ρ* expresses the meaning of electrical resistivity, frequency of EPT is denoted by *f*, the current density is replaced by *j_m_*, *τ_p_* is regarded as the duration of electro-pulsing treatment, *T* and *K* are absolute temperature and Boltzmann constant, respectively. Hence, based on Equation (1), it is seen that probability of the mobility of dislocations gets increased by electro-pulsing treatment, as shown in Figure 5d.

Another necessary condition to form plenty of sub-grains in S + EA sample is that there should be more new dislocations during electro-pulsing treatment. Also, it was reported that under the electric wind, not only the formation of sub-grain was promoted, but also the dislocation density was enhanced [22]. Consequently, the quantitative calculation of dislocation density is needed. In this research, the X-ray patterns profile method was applied by wide angle diffraction and the Rietveld software [23,24]. In this way, the dislocation density (*ρ*_0_) can be expressed as: (2)$$\rho_{0}=\frac{3\sqrt{2\;\pi }{\langle \varepsilon^{2}\rangle }^{1/2}}{D\;b}$$
where *ε* means the microstrain, *D* refers to the crystallite size, and *b* means the Burgers vector. Besides, the detailed values (listed in Table 4) of *ε* and *D* are calculated from Figure 3. Meanwhile, the calculated dislocation density is also given in Table 4. Apparently, the dislocation density of S + EA sample is obviously higher, which is consistent with the observation of Figure 4.

### 3.4. Quantitative Analysis of Higher UTS and Lower YS of S + EA Sample

Based on the results of tensile properties, it is worth noting that, although the ultimate strength of S + EA sample is higher, its yield strength is obviously lower than the T6 sample. This unusual phenomenon is bound to be related with the difference of microstructure between S + EA and T6 sample. 

The aforementioned results have indicated that there are many sub-grains in S + EA sample. However, this cell-like structure (dislocation cells) also exists in T6 and S sample. As shown in Figure 10, the content of dislocation cells in S sample is obviously higher than T6 sample, but is lower than S + EA sample (Figure 5d). Compared with S sample, the diameter of dislocation cells is obviously smaller in T6 sample and the corresponding content is also lower. Therefore, the glide distance of dislocations in S + EA sample is longer than T6 sample. As is reported in Reference [25], the glide distance of dislocations is the difference in size of dislocation cell between aging and solution treated. Compared with Figure 10 and Figure 5d, the glide distance of dislocations in S + EA and T6 sample is 240 nm and 180 nm, respectively.

Additionally, the existence of co-clusters can prevent the formation of dislocation tangle [26]. As is proved by APT, there exist a lot of Mg_2_(Si,Cu)_3_ clusters in S + EA sample. Thus, due to the difference of lattice constant between the clusters and matrix, there will exist elastic strain field around the clusters. Generally, this strain field can be considered as the obstacles of dislocation movement. In this way, the motion of dislocations will be restricted. In summary, it is reasonable to infer that compared with T6 sample, the dislocation entanglement in S + EA sample will be delayed due to the longer dislocation glide distance and prevention of dislocation pile-ups induced by clusters (Figure 11).

As can be seen from the above-mentioned discussion, the difference of yield strength is induced by precipitation strengthening mechanism. As revealed in Figure 11, assuming that there only exists precipitation strengthening in elastic deformation stage of both samples, the detailed quantitative calculations are as below.

In S + EA sample, the contribution (Δ*σ_p_*_1_) to YS is in relationship with volume fraction *f_v_* [27]:(3)$$\Delta \sigma_{p1}=M\frac{\alpha \;G\;b\sqrt{f_{v}}}{1.77\;r}$$
where the meaning and values of the *G*, *b* are shear modulus (about 25.9 GPa) and Burgers vector (0.286 nm), respectively; *M* is the average orientation factor (3.06), *r* refers to the mean radius of clusters. Whereas, in T6 sample, the corresponding strengthening mechanism is governed by bypassing mechanism (*r*/*b* > 15). Thus, the corresponding contribution (Δ*σ_p_*_2_) to yield strength is [26]:(4)$$\Delta \sigma_{p2}=M\frac{0.4\;G\;b}{\pi \sqrt{1-\nu }}\frac{ln\;(2r_{p}/b)}{\lambda_{P}}.$$

In Equation (4), *M* refers to the average orientation factor (3.06); *ν* means the poisson ratio (0.33); *r_p_* reflects average radius of Mg_2_Si phases, and *λ_p_* refers to average adjacent spacing of Mg_2_Si phases. As a result, the contribution of clusters to strength (S + EA) and the precipitate strengthening (T6) can be calculated and listed in Table 4. The calculated result indicates that the YS (yield strength) of T6 sample is 40.7 MPa higher than S + EA sample. Therefore, the lower YS of S + EA is attributed to the less effective clusters.

In fact, the dislocation strengthening should also be taken into consideration. In this way, the factual difference will be higher than 40.7 MPa. From the results of Figure 2a, the measured yield strength difference is about 93.4 MPa, which is higher than the calculated difference. This indicates that the dislocation strengthening is indeed delayed in S + EA sample.

On the other hand, during the plastic deformation stage, there will be many micro-voids in T6 sample and the sample is nearly fractured, due to the early entanglement of dislocations, as shown in Figure 11. Therein, the difference between ultimate strength is mainly related to a dislocation mechanism. As shown in Figure 11, we can assume there is only dislocation strengthening in the plastic stage of all samples. Thus, the increment from yield strength (Δ*σ_d_*) to ultimate strength induced by dislocation strengthening is [28]:(5)$$\Delta \sigma_{d}=M\alpha \;G\;b\rho^{1/2}.$$

Therefore, the strength increase induced by residual dislocation is listed in Table 4. It is found that the calculated difference is 66.8 MPa. Whereas, as has been discussed before, there exists dislocation entanglement in elastic stage of T6 sample. So, the contribution of dislocation strengthening in T6 sample to ultimate strength is actually lower. Consequently, the measured difference will be higher than 66.8 MPa. Based on the results of Table 2, the measured difference is about 100 MPa, which is higher than calculated value. Finally, this result also implies that the dislocation entanglement happens later in S + EA sample.

### 3.5. Superior Ductility of S + EA Sample Induced by Sub-Grains

Based on the results of Figure 4a–f, the grain size in S + EA sample is bimodal and new orientations appear in prior α-Al grains. Namely, the α-Al matrix is refined after electro-pulsing treatment. Consequently, higher energy is necessary for dislocations to go thorough the grain boundaries. And also, the corresponding plastic deformation will be more homogeneous. Thereupon, the ductility will be modified. In this viewpoint, the effects of geometrically necessary dislocations and the orientations of α-Al grains on ductility will be discussed.

As is reported, the sub-grains promoted the enhancement of ductility [29]. This indicates that the sub-grains in S + EA sample can be regarded as an effective structure. Huang [25] reminded that dislocations in metallic materials are classified into immobile and mobile dislocations. The boundaries of sub-grains (S + EA sample) and dislocation cell walls in T6 sample can be regarded as a high dislocation density area, whereas the central position of this structures is a low dislocation density area. During deformation, the mobile dislocation in the low dislocation density area will glide and this will induce break and collapses of initial cell boundaries (high dislocation density area). Generally, the plastic deformation (*ε_pl_*) related with mobile dislocation density (*ρ_m_*) is [25]:(6)$$\varepsilon_{pl}=\rho_{m}\;b\;l/M.$$

In Equation (6), *M* refers to Taylor factor, *b* means the Burgers vector, *l* is the abbreviation of the mean glide distance of mobile dislocations. The value of *ρ_m_* is 18% of the whole dislocation density [25]. Thus, the calculated ratio of *ε_pl_* (S + EA sample to T6 sample) is about 3.15. This result indicates that the sub-grain is a favorable factor for the superior ductility.

To accurately study the effects of new orientations on the plasticity of S + EA samples, the plastic deformation which is related to schmid factor needs to be quantitatively evaluated [30,31,32]:(7)$$\varepsilon_{pl}=\frac{\sigma_{Yield}\;M^{2}}{G}.$$

In Equation (7), *G* means the shear modulus, *M* refers to the average schmid factor, and *σ_yield_* represents the yield strength. From the statistical results of Figure 12a,c, the calculated value of schmid factor in S + EA and T6 sample are 0.455 and 0.402, respectively. In this way, the ratio of *ε_pl_* (S + EA sample to T6 sample) is 1.71, which demonstrates that the new orientations appear in S + EA sample is a favorable factor for the plasticity.

As mentioned in Section 3.1, the work-hardening curves of S + EA and T6 sample present different trends before their fracture. Therefore, in order to fully investigate this interesting phenomenon, SEM observations about the fracture morphology are carried out.

As shown in Figure 13, it is interestingly found that the dimples of T6 samples are large and homogeneous while the dimples of S + EA samples are heterogeneously distributed. In the fracture morphology of S + EA sample, densely distributed ultra-fine dimples were distributed inside the large dimples (Figure 13d). The completely different fracture morphology can be attributed to the prevention of crack propagation induced by sub-grains. Due to the large amount of sub-grains in S + EA sample, the crack propagation during fracture will be harder to pass thorough these obstacles. Therefore, there will exist local stress concentration and affected by this, the work-hardening rate will also increase before the fracture. Whereas, in T6 sample, there is no obstacle to crack propagation and so, the work-hardening rate will decrease continuously.

Figure 14 reveals the fracture process after tensile load is applied. As the work-hardening rate of S + EA increases before the fracture, the nucleation rate of local dimples will also increase rapidly (Figure 14b), making the dimples difficult to merge and grow [33]. Therein, a large amount of ultra-fine dimples is formed (Figure 13d). Whereas, the micro-voids can normally grow and merge in T6 sample. Consequently, the fracture of T6 sample illustrating a typical ductile rupture mode is shown in Figure 14a.

Additionally, as is mentioned in Reference [34], a relatively higher level of ductility can be reserved because of the prevention of dislocation pile-ups induced by clusters. That is to say, the formation of clusters is also favorable for the modification of ductility. In summary, high dislocation density does not always mean low ductility; other aspects such as the dislocation morphology and mobile dislocation density also need to be considered. The authors believe that this applies to other metallic materials as well.

## 4. Conclusions

The trade-off relationship between strength and ductility of 6061 alloy during aging process was broken by electro-pulsing treatment. Plenty of Mg_2_(Si,Cu)_3_ co-clusters appeared after electro-pulsing treatment. Meanwhile, a large amount of sub-grains is formed due to the special geometry shape of dislocations and the accelerated dislocation motion under the effects of electric pulsing. Besides, dislocation entanglement is delayed in S+EA sample due to the formation of Mg_2_(Si,Cu)_3_ and longer dislocation glide distance. More importantly, enhanced dislocation density induced by the formation of sub-grains offers a higher contribution to strength. Consequently, despite the lower yield strength, higher ultimate strength is still obtained. Furthermore, except for the favorable effects of co-clusters on ductility, the main reason for the higher ductility of S + EA sample is the formation of sub-grains. The detailed aspects can be classified as: (i) Higher mobile dislocation density and longer glide distance induce larger plastic deformation. (ii) New crystal orientations appear and higher schmid factors are obtained, which is favorable for slipping. (iii) Crack propagation is prevented by sub-grain boundaries. Finally, compared with T6 samples, the S + EA samples contain 78.9% higher ductility and maintain slightly higher strength within only 560 ms. In summary, EPT can be a more effective method for increasing ductility and toughness of 6061 alloys.

## Figures and Tables

**Figure 1 materials-12-01383-f001:**
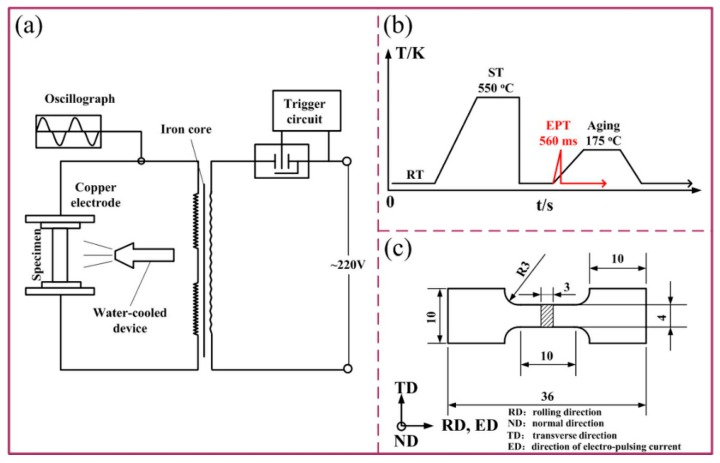
Schematic diagram showing (**a**) electro-pulsing machine, (**b**) treatment procedure, and (**c**) samples for tensile test.

**Figure 2 materials-12-01383-f002:**
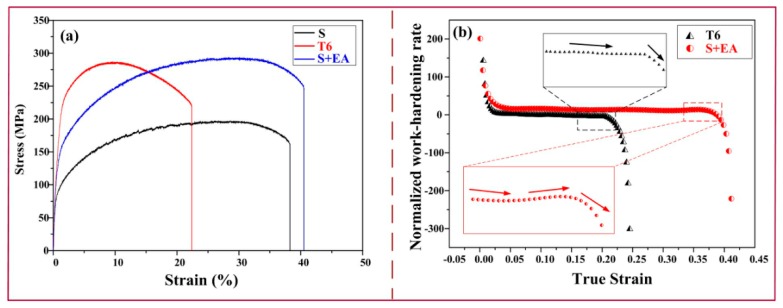
Properties of different samples: (**a**) Tensile properties and (**b**) normalized work-hardening rate curves.

**Figure 3 materials-12-01383-f003:**
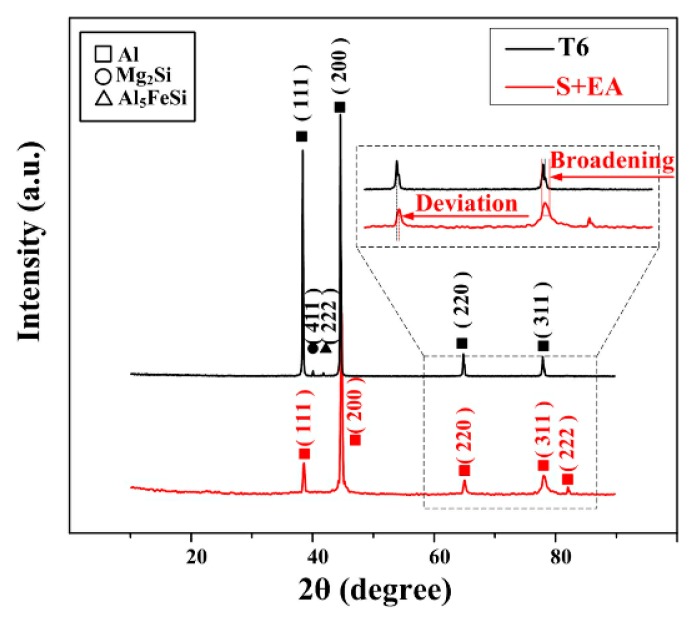
XRD spectrum of different samples.

**Figure 4 materials-12-01383-f004:**
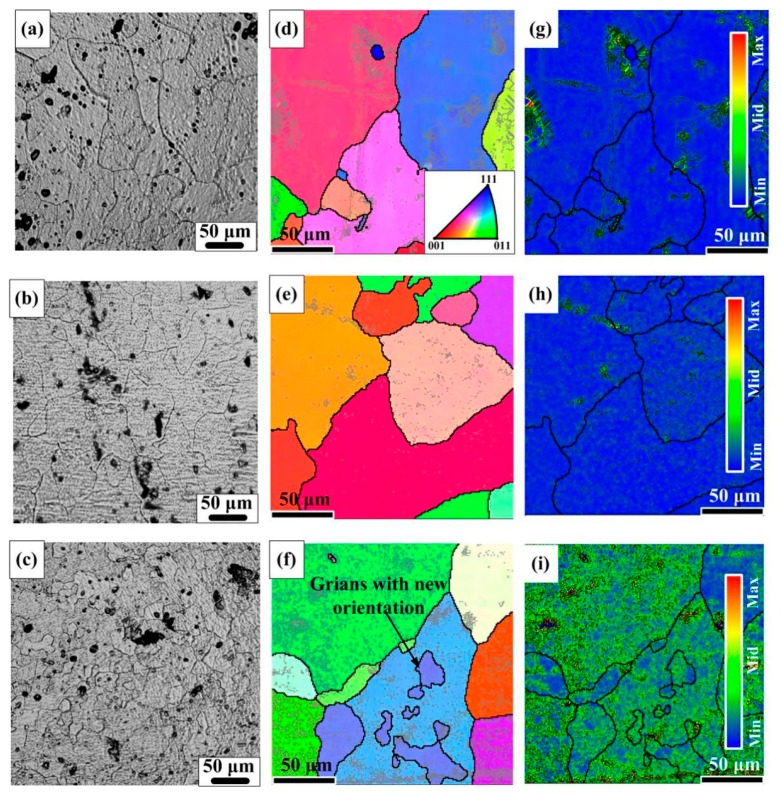
The optical images (**a**) S, (**b**) T6, (**c**) S + EA, and IPF (Inverse Pole Figure) images at rolled direction (RD), (**d**) S, (**e**) T6, (**f**) S + EA, (**g**–**i**) KAM distributions of (**d**–**f**), respectively. (Grid 200 × 200, step 1 µm, the mean angular deviation (MAD) of S sample, T6 sample, and S + EA sample are 0.62, 0.58, and 0.56, respectively, and the indexing percentage of S sample, T6 sample, and S + EA sample are 95.48%, 96.25%, and 97.35%, respectively).

**Figure 5 materials-12-01383-f005:**
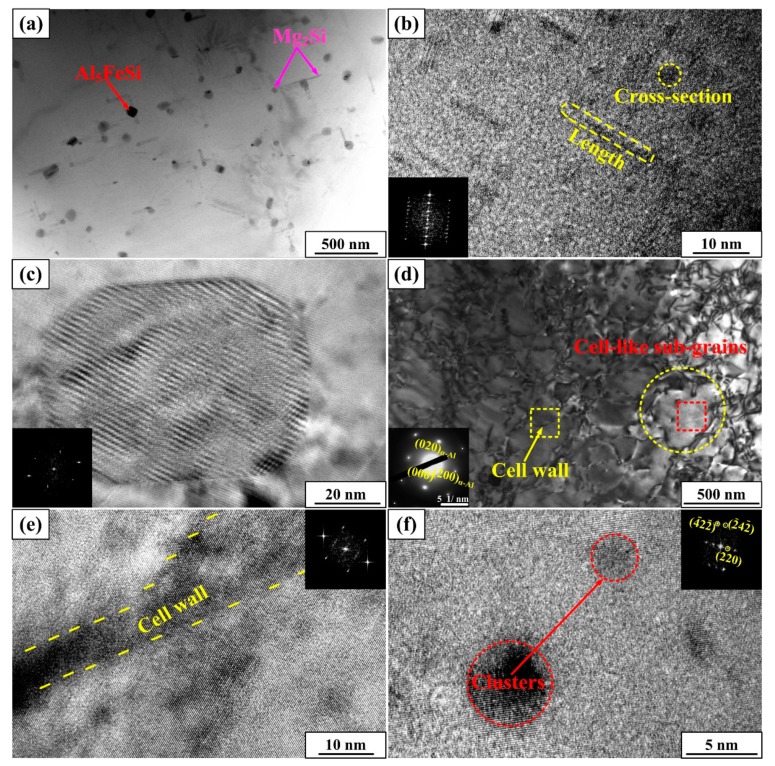
TEM images showing (**a**) precipitates in T6 sample; (**b**) high-resolution transmission electron microscopy (HRTEM) figure and fast Fourier-transform (FFT) of Mg_2_Si in (**a**); (**c**) HRTEM figure and FFT of Al_5_FeSi in (**a**); (**d**) Sub-grains in S + EA sample; (**e**) HRTEM and FFT of cell wall in (**d**); (**f**) HRTEM and FFT of clusters.

**Figure 6 materials-12-01383-f006:**
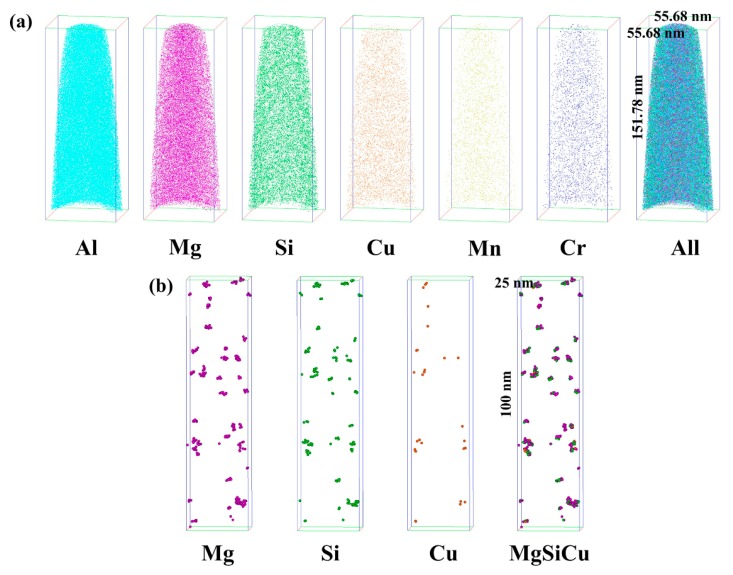
The atom probe tomograghy (APT) maps in S + EA sample: (**a**) Atom distribution maps of different elements in a cube of 55.68 nm × 55.68 nm × 151.78 nm, (**b**) cluster distribution maps of Mg, Si, Cu, and MgSiCu in selected cube of 25 nm × 25 nm × 100 nm.

**Figure 7 materials-12-01383-f007:**
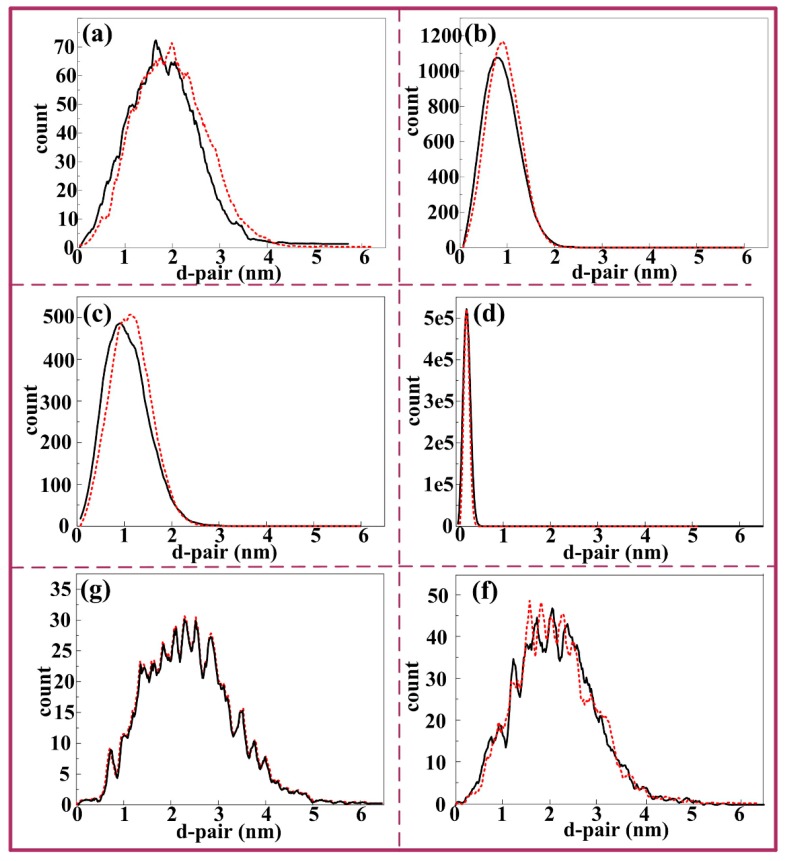
The nearest neighbor distribution of different elements in S + EA sample: (**a**–**f**) Cu, Mg, Si, Al, Mn, and Cr.

**Figure 8 materials-12-01383-f008:**
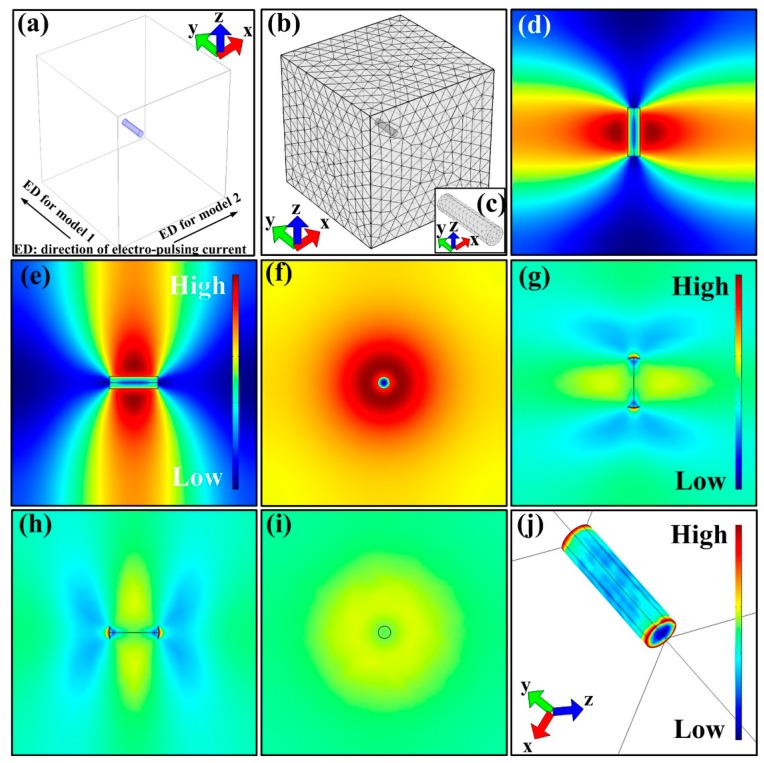
Simulation results of model 1: (**a**) Model construction, ED refers to electric current direction; (**b**) meshing of models; (**c**) an enlarged image of cylinder of dislocation in (b). (**d**–**f**) The distribution of temperature at XY, YZ, and XZ plane; (**g**–**i**) the distribution of current density at XY, YZ, and XZ plane; (**j**) thermal stress distribution.

**Figure 9 materials-12-01383-f009:**
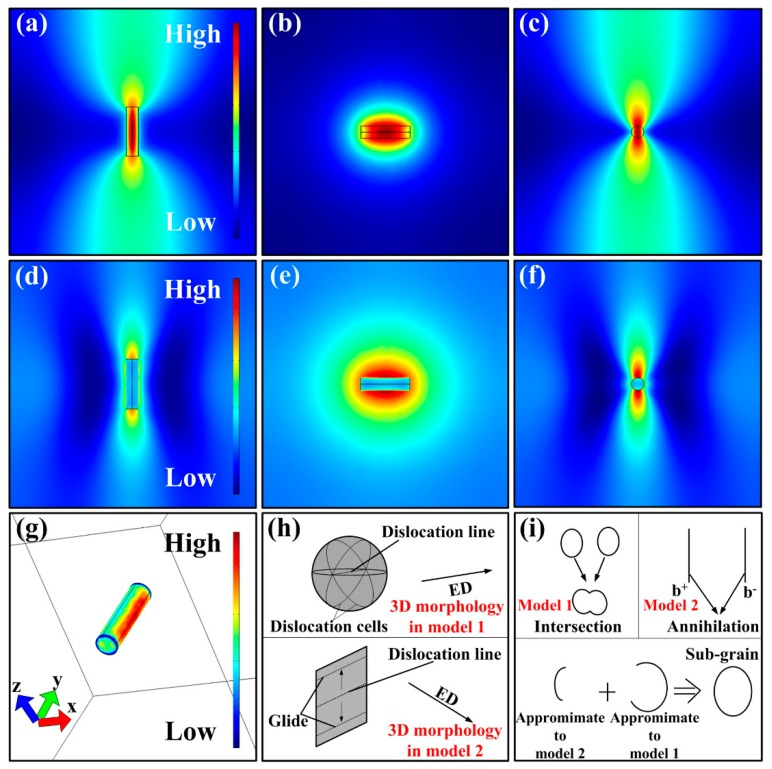
Simulation results of model 2: (**a**–**c**) Distribution of temperature at XY, YZ, and XZ plane; (**d**–**f**) distribution of current density at XY, YZ, and XZ plane; (**g**) distribution of thermal stress; (**h**) reveals the dislocation morphology after current was applied; (**i**) reflects dislocation motion in model 1, model 2, and the actual situation.

**Figure 10 materials-12-01383-f010:**
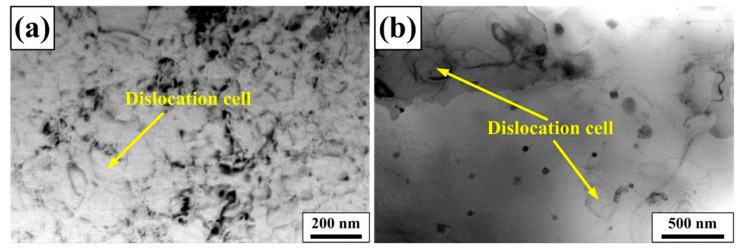
TEM observation showing morphology of dislocation cell of different samples: (**a**) S, (**b**) T6.

**Figure 11 materials-12-01383-f011:**
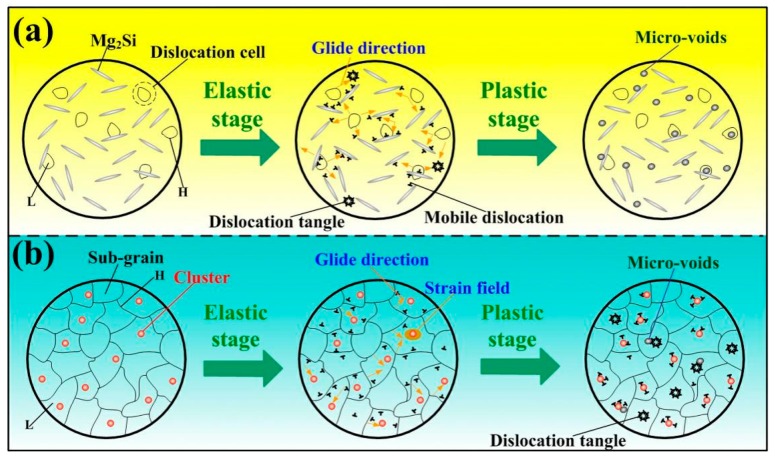
The schematic reveals the interaction of dislocation and precipitation in different states: (**a**) T6, (**b**) S + EA (L: low dislocation density, H: high dislocation density).

**Figure 12 materials-12-01383-f012:**
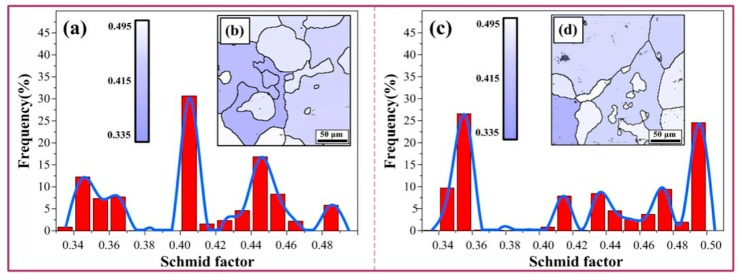
The schmid factor of different samples: (**a**,**b**) T6, (**c**,**d**) S + EA.

**Figure 13 materials-12-01383-f013:**
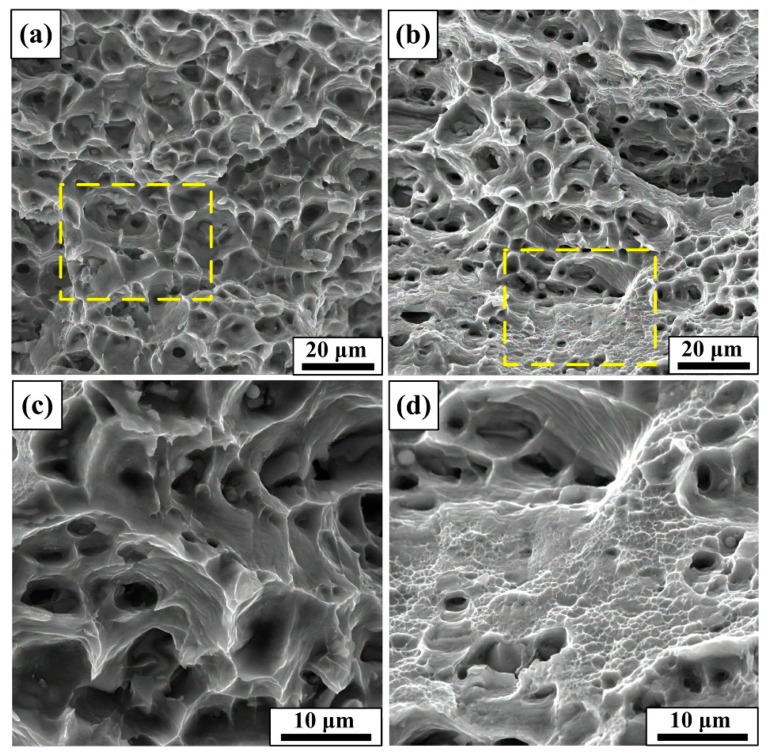
SEM observation showing the fracture morphologies in different states: (**a**,**c**) T6, (**b**,**d**) S + EA.

**Figure 14 materials-12-01383-f014:**
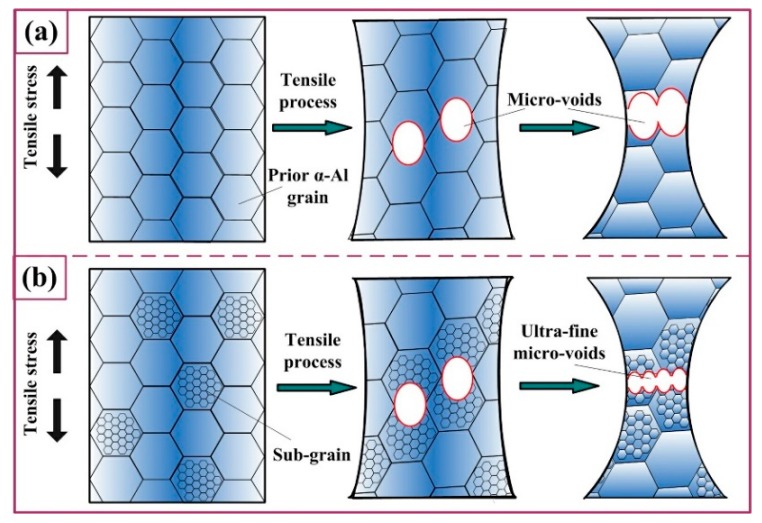
The schematic diagram of crack growth in different states: (**a**) T6, (**b**) S + EA.

**Table 1 materials-12-01383-t001:** The nominal compositions of 6061 sheet (wt.%).

Element	Mg	Si	Fe	Cu	Cr	Al
wt.%	0.88	0.64	0.43	0.24	0.13	Bal.

**Table 2 materials-12-01383-t002:** The tensile properties of 6061 samples in different states.

State	T6	S + EA	S
UTS (MPa)	286.1 ± 1.8	292.6 ± 2.1	196.5 ± 3.0
YS (MPa)	188.6 ± 1.4	95.2 ± 0.8	54.9 ± 1.2
El (%)	22.7 ± 1.2	40.6 ± 0.9	38.3 ± 1.4
SDB (MPa × %) = YS × El	6494.5 ± 2.2	11,879.6 ± 1.9	-

SDB refers to the strength-ductility balance (the SDB of S sample is not considered), El refers to elongation.

**Table 3 materials-12-01383-t003:** The chemical composition of each cluster counted by IVSA software.

*E&C_N_*	1^th^	2^th^	3^th^	…	56^th^	…	125^th^	…	166^th^
Cu Si Mg	11 22 22	13 24 25	20 38 40	… … …	14 26 25	… … …	76 163 157	… … …	10 19 21

Note: *E&C_N_* means the atomic amount of element (Mg, Si and Cu) in the *i^th^* cluster.

**Table 4 materials-12-01383-t004:** Necessary parameters for the calculation of dislocation density and the detailed contribution to yield strength.

State	D (Å)	*ε* (×10^4^)	*ρ* (/m^2^)	Δ*σ_d_* (MPa)	Δ*σ_p1_*/Δ*σ_p2_* (MPa)
S + EA	486.2	435.7	2.6 × 10^15^	179.1	44.5
T6	1115.6	466.3	1.1 × 10^15^	112.3	85.2

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
