# Peer review of "Simultaneously Enhanced Strength and Ductility of Al–Mg–Si Alloys during Aging Process Induced by Electro-Pulsing Treatment"

_materials, 2019, doi:10.3390/ma12091383_

Round 1
Reviewer 1 Report
The manuscript is devoted to the study of the microstructure and mechanical properties of the 6061 alloy after electro-pulsing treatment. Interesting results were obtained regarding microstructural characterization and the achievement of high plasticity. However, work can be significantly improved by considering the following comments:
1. Usually, the alloy in the T6 state has higher values of yield strength and tensile strength (270 and 310 MPa, respectively) than those obtained in this work. Therefore, electro-impulse treatment does not increase strength compared to T6. In addition, on page 4, the authors write that: “In general, as is seen in Tab. 2, the quality index of S+EA sample is 44 MPa higher than T6 sample, which indicates that electro-pulsing treatment has more obvious advantages on the modification of 6061 alloys.” However, this quality index is descriptive of the true tensile properties of a casting and does not take into account the yield strength, which is actually almost three times lower than the usual value. Therefore, to assert that electro-pulsing treatment has more obvious advantage is not correct. In addition, it would be desirable to compare the mechanical properties (and microstructure) of the alloy after electropulsing treatment not only with the T6 state, but also with the state after natural aging and a short artificial one based on literature data. (for example, Aytekin Polat, Mustafa Avsar, Fahrettin Ozturk, Effects of the artificial-aging temperature and time on the mechanical properties and springback behavior of AA6061, Materials and technology 49 (2015) 4, 487–493).
2. In Materials and Methods, it should be indicated how the elongation was measured. Using an extensometer or by crosshead displacement?
3. In line 74:“Solution treatment” should be written instead of “S treatment”. Besides, electro-pulsing treatment is called as EA. Why?
4. On page 4 authors write: ”The aforementioned results indicate that mechanical properties of 6061 alloys can be remarkably enhanced by electro-pulsing treatment.” And in conclusions on page 15: “In summary, EPT can be an effective toughening and strengthening method for 6061 alloys.” I would say that this method is more effective for increasing ductility and toughness, not strength.
5. In the text, there are errors, words written together. On page 1 in line 29: alloysare, on page 3 in line 87: appliedfor, on page 6 in line 167: indicatesthat.
6. In Fig. 2 (a) “stress” should be written instead of “strength”.
7. Fig. 10 and discussion should be clarified. The figure shows the TEM microstructures in quenching (S) and T6 states, but in discussion, the T6 and “S+EA” samples are compared. In addition, unlike the text, the Fig. 10 (b) shows that the distance between the cells is large and their number is small, so the size of the cells should not have a significant impact on strength.
8. Fig. 11 and subsequent discussion are not clear. On page 11 authors write: “Additionally, the existence of co-clusters can prevent the formation of dislocation tangles [26]”. What does it mean? Also: “As revealed in Fig. 11, assuming that there only exists precipitation strengthening in elastic deformation stage of both samples, the detailed quantitative calculations are as below. “ (line 290) And “In fact, the dislocation strengthening also occurs during elastic stage of T6 sample.” (line 304). Why does strengthening occur during elastic deformation?? In addition, in Fig. 11 “dislocation” should be written instead of “dislogation”.
9. Formula (3) has no reference. The contribution of clusters to YS has an index “Orowan”, although the Orowan mechanism is related to the dislocation bowing between particles, and the clusters must be cut. Besides, formula (5) has a different spelling in [27].
10. On page 12 authors write: “The calculated result indicates that the YS (yield strength) of S+EA sample is 43.0 MPa higher than T6 sample.“ But in table, the value of S+EA sample is lower than T6 sample. And later: “This indicates that the dislocation strengthening is indeed delayed in S+EA sample.” In general, in my opinion, the lower value of the yield strength after electroimpulse treatment is due to the fact that the precipitated clusters are less effective reinforcers than strengthening phases in the T6 state.
Author Response
Dear Editors and Reviewers:
Thank you for your letter and for the reviewer's comments concerning our manuscript entitled "Simultaneously enhanced strength and ductility of Al-Mg-Si alloys during aging process induced by electro-pulsing treatment" submitted to Materials (No: materials-487616). Those comments are all valuable and very helpful for revising and improving our paper, as well as the important guiding significance to our researches. We have studied comments carefully and have made correction which we hope meet with approval. Revised portion are marked in yellow in the paper. The following lists our point-to-point responses to the reviewer's comments and the changes made in the revised manuscript.
Responds to the reviewer's comments:
Reviewer #1:
1. Usually, the alloy in the T6 state has higher values of yield strength and tensile strength (270 and 310 MPa, respectively) than those obtained in this work. Therefore, electro-impulse treatment does not increase strength compared to T6. In addition, on page 4, the authors write that:“In general, as is seen in Tab. 2, the quality index of S+EA sample is 44 MPa higher than T6 sample, which indicates that electro-pulsing treatment has more obvious advantages on the modification of 6061 alloys.” However, this quality index is descriptive of the true tensile properties of a casting and does not take into account the yield strength, which is actually almost three times lower than the usual value. Therefore, to assert that electro-pulsing treatment has more obvious advantage is not correct. In addition, it would be desirable to compare the mechanical properties (and microstructure) of the alloy after electropulsing treatment not only with the T6 state, but also with the state after natural aging and a short artificial one based on literature data. (for example, Aytekin Polat, Mustafa Avsar, Fahrettin Ozturk, Effects of the artificial-aging temperature and time on the mechanical properties and springback behavior of AA6061, Materials and technology 49 (2015) 4, 487–493).
Response: Thanks very much for the reviewer’s comments. Based on a large amount of data from tensile test, the yield and ultimate strength are indeed the same as the detailed values given in the manuscript. The difference between our values and the values proposed by reviewers may be related with the different chemical composition, initial state and processing conditions of this kind alloy. Besides, for the quality index, as the reviewer suggested, this index is truly inappropriate to interpret the comprehensive mechanical properties of 6061 alloy. We are sorry for our negligence, and the corresponding index has been changed into the strength-ductility balance (the products of ultimate tensile strength and ductility). Also, the reference 14 detailed interpretation in text has also been corrected. Moreover, for the comparison between S+EA sample and S+EA+artificial sample, the detailed results has been published in our previous work entitled " Superior mechanical properties induced by the interaction between dislocations and precipitates in the electro-pulsing treated Al-Mg-Si alloys, Mater. Sci. Eng. A, 735 (2018) pp 154-161". Compared with T6 sample, the S+EA+artificial sample has a huge increase in both yield strength and ultimate strength, besides, the ductility can maintain the same level of T6 sample. On the other hand, for the comparison of S+EA sample and S+EA+natural aging, as the reviewer suggested, it’s very important to reveal the advantage and stability of this electropulsing treatment. Whereas, due to the limited time for the revision of manuscript, and the aim of this manuscript is mainly to demonstrate the enhancement of ductility induced by electro-pulsing, this comparison will be displayed in our future research. Thanks again for the reviewer’s constructive suggestions.
2. In Materials and Methods, it should be indicated how the elongation was measured. Using an extensometer or by crosshead displacement?
Response: Thanks for the reviewer’s reminding and we are sorry for this negligence. The elongation in this manuscript was measured by a BCX5000 extensometer matched to the MTS-810 tensile test equipment. This interpretation has been added to section 2.
3. In line 74:“Solution treatment” should be written instead of “S treatment”. Besides, electro-pulsing treatment is called as EA. Why?
Response: Thanks for the reviewer’s comments. We are sorry for our mistakes. We have replaced the term " S treatment" by " Solution treatment" . Besides, the term " EA" is the abbreviation of " electro-pulsing treatment followed with air cooling" and we have mentioned this issue in the section 2 of the manuscript.
4. On page 4 authors write: “The aforementioned results indicate that mechanical properties of 6061 alloys can be remarkably enhanced by electro-pulsing treatment.” And in conclusions on page 15: “In summary, EPT can be an effective toughening and strengthening method for 6061 alloys.” I would say that this method is more effective for increasing ductility and toughness, not strength.
Response: Thanks for your comments. In fact, as a rapid treating method, many researches have shown that electro-pulsing treatment is an effective strengthening and toughening method. However, as the reviewer suggested, in this manuscript, this method is more effective for increasing ductility. We are sorry for this inappropriate interpretation and the detailed section in conclusion has been modified.
5. In the text, there are errors, words written together. On page 1 in line 29: alloysare, on page 3 in line 87: appliedfor, on page 6 in line 167: indicatesthat.
Response: Thanks for the reminding and we are very sorry for the word mistakes. These mistakes have been corrected and we make sure not to make such mistakes in the future.
6. In Fig. 2 (a) “stress” should be written instead of “strength”.
Response: Thanks for the reviewer’s suggestion and the term "stress" has been replaced by "strength".
7. Fig. 10 and discussion should be clarified. The figure shows the TEM microstructures in quenching (S) and T6 states, but in discussion, the T6 and “S+EA” samples are compared. In addition, unlike the text, the Fig. 10 (b) shows that the distance between the cells is large and their number is small, so the size of the cells should not have a significant impact on strength.
Response: Thanks for the reviewer’s suggestions. As the reviewer suggested, this paragraph is indeed not clearly described. We have corrected this section and this discussion has also been clarified. Besides, the diameter of dislocation cells is to reveal the effects on ductility of 6061 alloy, not the strength.
8. Fig. 11 and subsequent discussion are not clear. On page 11 authors write: “Additionally, the existence of co-clusters can prevent the formation of dislocation tangles [26]”. What does it mean? Also: “As revealed in Fig. 11, assumi ng that there only exists precipitation strengthening in elastic deformation stage of both samples, the detailed quantitative calculations are as below. “ (line 290) And “In fact, the dislocation strengthening also occurs during elastic stage of T6 sample.” (line 304). Why does strengthening occur during elastic deformation?? In addition, in Fig. 11 “dislocation” should be written instead of “dislogation”.
Response: Thanks for the reviewer’s comments. The detailed prevent effects on formation of dislocation tangles induced by clusters means that the strain field around the clusters can be considered as the obstacles of the dislocation movement. In this way, the dislocation entanglement will be delayed. This interpretation has been modified in this manuscript. Besides, from fig. 2b, work-hardening which is related with dislocation motion has occurred at elastic stage. So, in elastic stage, the strengthening mechanism can be selected as an index to evaluate the yield strength. Whereas, strengthening happens in elastic stage is indeed inappropriate and we have corrected the corresponding interpretation in the manuscript. Finally, following the reviewer’s comments, the term "dislogation" has been replaced by "dislocation". We will be careful not to make such mistakes in the future.
9. Formula (3) has no reference. The contribution of clusters to YS has an index “Orowan”, although the Orowan mechanism is related to the dislocation bowing between particles, and the clusters must be cut. Besides, formula (5) has a different spelling in [27].
Response: Thanks for your comments. We have added a new reference to formula (3). Besides, the term "the contribution of clusters to YS" has been replaced by "the contribution to YS ". As for the reference to formula (5), we have added the accurate reference to replace the reference [27].
10. On page 12 authors write: “The calculated result indicates that the YS (yield strength) of S+EA sample is 43.0 MPa higher than T6 sample.“But in table, the value of S+EA sample is lower than T6 sample. And later: “This indicates that the dislocation strengthening is indeed delayed in S+EA sample.”In general, in my opinion, the lower value of the yield strength after electroimpulse treatment is due to the fact that the precipitated clusters are less effective reinforcers than strengthening phases in the T6 state.
Response: Thanks for the comments. We are truly sorry for this mistake. From Table. 4, the calculated contribution to yield strength induced by dislocation strengthening in T6 sample is 43 MPa higher than the S+EA sample. Whereas, in the manuscript, we reversed this quantitative relationship. So, we have corrected the corresponding interpretation. What’s more, the authors believe the reviewer’s viewpoint that the precipitated clusters are less effective, and this can directly demonstrate the reason of lower YS of S+EA sample. Thus, we also added this interpretation to section 3.4. The supplementary calculation of dislocation strengthening in both elastic and plastic stage is only to point that the dislocation entanglement is indeed delayed in S+EA sample from a quantitatively viewpoint.
All the modifications are highlighted using yellow background in the revised manuscript.
We tried our best to improve the manuscript and made some changes in the manuscript, however, if the reviewers are not satisfied with that, we could make the further revision. Finally, we are grateful to the reviewer for his (her) critical reading of this manuscript and constructive suggestions. All the questions are important and helpful to improve our manuscript. We hope that the changes incorporated in the revised manuscript could be acceptable.
Best regards
Xiaofeng Xu.
Reviewer 2 Report
Overview and general recommendation
The presented study gives an overview about the influence of electro-pulsing treatment of an Al-Mg-Si alloy compared to conventional aging treatment in order to improve strength and ductility simultaneously. The authors applied different methods to investigate the microstructural changes as well as to discuss responsible mechanisms. I found the paper to be overall well written and much of it to be well described. I felt confident that the authors performed careful experiments and investigations. However, the analysis of the EBSD measurements, the basis of their discussion must be improved before publication. Further, it should be considered, that the aim of improving strength by electro-pulsing treatment was only partially fulfilled. The causes for this were discussed, but the resulting possibly disadvantageous differences to the T6 state were ignored. Therefore, I recommend major revision. Please find my remarks in detail below.
Major comments 1. Line 47 to 68: Please check references carefully! Ma does not match with Ref. 10, Wang not with Ref. 12. 2. Materials and Methods: How was the used material produced? Which condition was used? Please add further details and sheet thickness. 3. Line 75: Which kind of cooling was used? Please add. 4. Figure 1: Please define abbreviations of the axis designations. Define ED in (c). 5. Line 118: Table caption does not refer to this table. 6. The results of the EBSD measurements are the basis of the discussion, which mechanisms are responsible for the microstructural evolution. A measurement area of about 200 x 200 μm for microstructures exhibiting an average grain size of about 50 μm is insufficient. Number of measured grains is lower than 20 and seems not be representative. Furthermore, grey lines, which should represent grain boundaries less than 2°, are not visible (g), (h), (i). Please use uniform maximum axis values for better comparison. New orientations, like mentioned in Line 164, were not described here, when results are shown. But please consider, that it is not sufficient to measure 20 grains and conclude that here are differences in orientations for several conditions. In line 322 the grain size for S+EA is described as bimodal. Here, too, it is important to note that a bimodal grain size cannot be estimated from 20 grains. Please add grain size measurements for all conditions. 7. Figure 6a: Distribution seems to be homogenously. I cannot comprehend the conclusion in lines 180 and 181. Comparison to T6-condition is missing, which would be useful for clarification that no clusters form after T6 treatment. 8. Figure 7: Nearest neighbor distribution of several elements are shown, but not described within the text. Please add a short summary whether the lines coincide or not. For 7(f) black line seems to deviate to the right. Please add the effect of these behavior. 9. Line 391: It is concluded that EPT is effective for strengthening 6061 alloy. But please consider that the YS was lower compared to the T6-treated condition. The responsible mechanisms were discussed before. However, if the lower YS offers disadvantages, for example plastic flow starts earlier, was ignored. Please add short comment to this regard. Minor comments 1. Line 29: blank space between alloysare
2. Line 35: through instead of thorough?
3. Line 36: Higher amount of stacking faults or higher stacking fault energy? Please specify.
4. Line 38: Equal channel angular extrusion = ECAE
5. Line 44: less effective with regard to? Please add.
6. Line 60 to 67: No capital letter for Electron in Electron scanning microscope, and Transmission, and Atom, etc.
7. Generally: Please do not use abbreviations for is or are like you have done in line 67, 105, 241,…
8. Line 74: Please define S, when used for the first time.
9. Line 82: No capital letter for Electro
10. Line 84: Please use tensile tests instead of mechanical properties tests
11. Line 85: Please use each condition instead of tensile test in every state
12. Line 87: blank space between appliedfor
13. Generally: UTS = Ultimate tensile strength (instead of ultimate strength)
14. Generally: Use uniform figure captions: (a) XXXX and (b) XXXX or XXXX (a) and XXXX (b)
15. Line 115: Please add Q to quality index
16. Line 133: Please define FWHM
17. Line 134: precipitates instead of precipitations
18. Line 143: blank space between than2°
19. Line 158: Need-like = needle-like?
20. Line 159: Please define FFT
21. Line 167: blank space between indicatesthat
22. Figure 8: Cylinder of dislocation hard to see in (a) and (b)
23. Equation (1) and (2): Both equations contain ρ but with different meaning. Is it possible to differentiate, for example using a subscripted character?
24. Figure 10 and line 272 to 278: Caption of Fig. 10 suggests that (a) refers to S. But the text in line 272 to 278 is aimed at S+EA. Please check whether S and T6 or S+EA and T6 are compared.
25. Figure 11: Please define L and H. Yellow writing is hard to recognize. Maybe usage of another color is useful.
26. Equation (6): Please define l.
27. Line 363 to 373: References 32, 33 and 26 seem not fit the topic described. Please check!
Author Response
Dear Editors and Reviewers:
Thank you for your letter and for the reviewer's comments concerning our manuscript entitled "Simultaneously enhanced strength and ductility of Al-Mg-Si alloys during aging process induced by electro-pulsing treatment" submitted to Materials (No: materials-487616). Those comments are all valuable and very helpful for revising and improving our paper, as well as the important guiding significance to our researches. We have studied comments carefully and have made correction which we hope meet with approval. Revised portion are marked in yellow in the paper. The following lists our point-to-point responses to the reviewer's comments and the changes made in the revised manuscript.
Responds to the reviewer's comments:
Reviewer #2:
Major comments
1. Line 47 to 68: Please check references carefully! Ma does not match with Ref. 10, Wang not with Ref. 12.
Response: Thanks for your suggestion. We are very sorry for our negligence. The Ref.10 and the Ref.12 are inverted. “In Ma’s research” has been changed by “In Zhao’s research”, and is referred to new Ref.10, and the work in Wang is referred to the new Ref.12 throughout the manuscript to keep consistent.
2. Materials and Methods: How was the used material produced? Which condition was used? Please add further details and sheet thickness.
Response: Following your suggestion. The commercial AA6061 was rolled and annealed, the sheet thickness was 10 mm. We have added this interpretation to section 2.
3. Line 75: Which kind of cooling was used? Please add.
Response: Thanks for your comments. We are very sorry for our negligence. The cooling after electro-pulsing treatment is air-cooling. We have added this interpretation to Line 75.
4. Figure 1: Please define abbreviations of the axis designations. Define ED in (c).
Response: Thanks for your comments. We are very sorry for our negligence. RD is for rolling direction, ND is for normal direction, TD is for transverse direction, ED is for the direction of electro-pulsing current. We have added this interpretation to the revised manuscript.
5. Line 118: Table caption does not refer to this table.
Response: We are sorry for our mistake. We have replaced the term " Table 2. The nominal compositions of 6061 sheet (wt.%)." with " Table 2. The mechanical properties of 6061 samples in different states " throughout the manuscript to keep consistent.
6. The results of the EBSD measurements are the basis of the discussion, which mechanisms are responsible for the microstructural evolution. A measurement area of about 200 x 200 μm for microstructures exhibiting an average grain size of about 50 μm is insufficient. Number of measured grains is lower than 20 and seems not be representative. Furthermore, grey lines, which should represent grain boundaries less than 2°, are not visible (g), (h), (i). Please use uniform maximum axis values for better comparison. New orientations, like mentioned in Line 164, were not described here, when results are shown. But please consider, that it is not sufficient to measure 20 grains and conclude that here are differences in orientations for several conditions. In line 322 the grain size for S+EA is described as bimodal. Here, too, it is important to note that a bimodal grain size cannot be estimated from 20 grains. Please add grain size measurements for all conditions.
Response: Thanks for the reviewer’s comments. A measurement of 200 × 200 μm is only to clearly show the existence of grains with new orientations at a high magnification in S+EA sample. Besides, we make sure that all the phenomenon measured by EBSD is universal and we choose the most representative one in the manuscript. Unfortunately, we haven’t collected EBSD figures containing more grains and due to the extremely limited time for manuscript revision, we also haven’t enough time for further EBSD test. Moreover, as the reviewer suggested, the EBSD figures with gray lines have been replaced with the ones containing black lines. Meanwhile, the interpretation about appearance of small grains with new orientations has been added to the corresponding section. Finally, from our statistical results by IPP software, the grain size in S+EA sample is indeed bimodal (large grain and interior small grain with new orientation). Whereas, following the reviewer’s comments, the EBSD figures at a high magnification are not representative. Therefore, the detailed grain size will not be discussed in the revision.
7. Figure 6a: Distribution seems to be homogenously. I cannot comprehend the conclusion in lines 180 and 181. Comparison to T6-condition is missing, which would be useful for clarification that no clusters form after T6 treatment.
Response: Thanks for your comments. Due to the extremely small size of clusters, the equal concentration morphology of clusters can’t be obtained by IVSA software. Whereas, based on the results of fig. 7, the enrichment of Mg, Si and Cu have been quantitatively verified. Therein, following the reviewer’s suggestion, the inappropriate conclusions in lines 180 and 181 have been corrected. Besides, for the T6 sample, from the results of HRTEM figure (fig. 5b), there only exist Mg2Si and Al5FeSi phases. Besides, due to the adequate time for the growth, the authors believe that there aren’t clusters exist in T6 sample. In order to avoid misunderstanding about this issue, we have added the corresponding interpretation to the manuscript.
8. Figure 7: Nearest neighbor distribution of several elements are shown, but not described within the text. Please add a short summary whether the lines coincide or not. For 7(f) black line seems to deviate to the right. Please add the effect of these behavior.
Response: Thanks for your comments. We are very sorry for our negligence. In Fig. 7a-c, from the nearest distribution of Cu, Mg, Si, the black line deviates to the left of the red line obviously. As is seen in Fig. 7, the red line represents the distribution curves of the distance between the two nearest neighbors if the atoms are randomly distributed, while the black line represents the distance distribution curves between the two nearest neighbors in the actual test data collection. After carefully confirmation, if the black line deviates to the left of the red line, it indicates that the atomic spacing of the actual test data is less than the random distribution and this element is enriched. If not (whether the black line coincides with red line or black line deviates to the right), it means that the actual test data is consistent with the random distribution, and the element is randomly distributed. Based on this, in Fig. 7f, the nearest distribution of Cr, which the black line deviates to the left also means the element is randomly distributed. We have added this interpretation to the revised manuscript.
9. Line 391: It is concluded that EPT is effective for strengthening 6061 alloy. But please consider that the YS was lower compared to the T6-treated condition. The responsible mechanisms were discussed before. However, if the lower YS offers disadvantages, for example plastic flow starts earlier, was ignored. Please add short comment to this regard.
Response: Thanks for your comments. As the reviewer suggested, the lower YS is indeed an unfavorable factor for Al-Mg-Si alloys to be applied in the industry. Our aim is to enhance both the strength and ductility of Al-Mg-Si alloys by electro-pulsing treatment. Our previous work entitled " Superior mechanical properties induced by the interaction between dislocations and precipitates in the electro-pulsing treated Al-Mg-Si alloys, Mater. Sci. Eng. A, 735 (2018) pp 154-161" has indicate that the superior strength of 6061 alloy can be obtained by electric current at a high voltage and low duration, meanwhile, the ductility can maintain the same level as T6 sample. Whereas, in this work, a Al-Mg-Si alloy with superior ductility is obtained by electric current at a low voltage and high duration, meanwhile, even though the YS is lower than T6 sample, the UTS is higher. Both the two works show that the electro-pulsing treatment has the potential to improve the strength and ductility at the same time. Therefore, by our further research in future, we believe that all the YS, UTS and ductility of 6061 alloy can be modified by electro-pulsing treatment. In this way, some disadvantages mentioned by the reviewer, for example, earlier plastic flow can be avoided. Finally, sincerely thanks the reviewer’s constructive comments.
Minor comments
1. Line 29: blank space between alloysare
Response: We are sorry for our mistake. We have replaced the term " alloysare" with " alloys are " throughout the manuscript to keep consistent.
2. Line 35: through instead of thorough?
Response: We are sorry for our mistake. We have replaced the term "thorough" with " through" throughout the manuscript to keep consistent.
3. Line 36: Higher amount of stacking faults or higher stacking fault energy? Please specify.
Response: Thanks for your comments. We are very sorry for our negligence. The correct statement is higher stacking fault energy. We have added this interpretation to the revised manuscript.
4. Line 38: Equal channel angular extrusion = ECAE
Response: Thanks for your comments. We are very sorry for our negligence. The Equal channel angular extrusion should be truly abbreviated as ECAE. We have added this interpretation to the revised manuscript.
5. Line 44: less effective with regard to? Please add.
Response: Thanks for your comments. We are very sorry for our negligence. The traditional aging treatment and the aforementioned refinement methods are not only less effective because of complicated process, but also time-wasting for Al-Mg-Si alloys. We have added this interpretation to the revised manuscript.
6. Line 60 to 67: No capital letter for Electron in Electron scanning microscope, and Transmission, and Atom, etc.
Response: Thanks for your comments. We are very sorry for our negligence. We have added this interpretation to the revised manuscript.
7. Generally: Please do not use abbreviations for is or are like you have done in line 67, 105, 241,…
Response: We are sorry for our mistake. We have added this interpretation to the revised manuscript. We will be careful not to make such mistakes in the future.
8. Line 74: Please define S, when used for the first time.
Response: We are sorry for our mistake. We have replaced the term "S" with " Solution" throughout the manuscript to keep consistent.
9. Line 82: No capital letter for Electro
Response: We are sorry for our mistake. We have corrected this term to the revised manuscript.
10. Line 84: Please use tensile tests instead of mechanical properties tests
Response: We are sorry for our mistake. We have replaced the term "mechanical properties tests" with "tensile tests" throughout the manuscript to keep consistent.
11. Line 85: Please use each condition instead of tensile test in every state
Response: We are sorry for our mistake. We have replaced the term "in every state" with "in each condition" throughout the manuscript to keep consistent.
12. Line 87: blank space between appliedfor
Response: We are sorry for our mistake. We have replaced the term "appliedfor" with "applied for" throughout the manuscript to keep consistent.
13. Generally: UTS = Ultimate tensile strength (instead of ultimate strength)
Response: We are sorry for our mistake. We have replaced the term "UTS (ultimate strength)" with "UTS (ultimate tensile strength)" throughout the manuscript to keep consistent.
14. Generally: Use uniform figure captions: (a) XXXX and (b) XXXX or XXXX (a) and XXXX (b)
Response: Thanks for your comments. We are very sorry for our negligence. We have corrected this interpretation to the revised manuscript.
15. Line 115: Please add Q to quality index
Response: Thanks for your comments. The quality index in Line 115 should be modified as Q, but as the suggestions of reviewer 1, the quality index is descriptive of the true tensile properties of a casting, the statement is not appropriate. So we have replaced the term "quality index" with "strength-ductility balance" throughout the manuscript to keep consistent.
16. Line 133: Please define FWHM
Response: Thanks for your comments. We are very sorry for our negligence. FWHM is the abbreviation of Full Width Half Maximum. We have added this interpretation to the revised manuscript.
17. Line 134: precipitates instead of precipitations
Response: We are sorry for our mistake. We have replaced the term "precipitations" with "precipitates" throughout the manuscript to keep consistent.
18. Line 143: blank space between than2°
Response: We are sorry for our mistake. We have replaced the term "than2°" with "than 2°" throughout the manuscript to keep consistent.
19. Line 158: Need-like = needle-like?
Response: We are sorry for our mistake. We have replaced the term "need-like" with "needle-like" throughout the manuscript to keep consistent.
20. Line 159: Please define FFT
Response: Thanks for your comments. We are very sorry for our negligence. FFT is the abbreviation of Fast Fourier-Transform. We have added this interpretation to the revised manuscript.
21. Line 167: blank space between indicatesthat
Response: We are sorry for our mistake. We have replaced the term "indicatesthat" with "indicates that" throughout the manuscript to keep consistent.
22. Figure 8: Cylinder of dislocation hard to see in (a) and (b)
Response: Thanks for your comments. We are very sorry for our negligence. We have modified the color of cylinder of dislocation in the revised manuscript.
23. Equation (1) and (2): Both equations contain ρ but with different meaning. Is it possible to differentiate, for example using a subscripted character?
Response: Thanks for your comments. We are very sorry for our negligence. We have replaced the term "ρ" with "ρ0" in the Equation (2) throughout the manuscript to keep consistent.
24. Figure 10 and line 272 to 278: Caption of Fig. 10 suggests that (a) refers to S. But the text in line 272 to 278 is aimed at S+EA. Please check whether S and T6 or S+EA and T6 are compared.
Response: We are sorry for our mistake. We have replaced the term in Line 272 to 278 "S+EA" with "S" throughout the manuscript to keep consistent.
25. Figure 11: Please define L and H. Yellow writing is hard to recognize. Maybe usage of another color is useful.
Response: Thanks for your comments. We are very sorry for our negligence. L is the abbreviation of low dislocation density, H is the abbreviation of high dislocation density. We have added this interpretation to the caption of Fig. 11 in the revised manuscript. As for the yellow writing, we have changed the color of the schematic in the revised manuscript.
26. Equation (6): Please define l.
Response: Thanks for your comments. We are very sorry for our negligence. The l is the abbreviation of the mean glide distance of mobile dislocations. We have added this interpretation to the revised manuscript.
27. Line 363 to 373: References 32, 33 and 26 seem not fit the topic described. Please check!
Response: Thanks for your comments. We are very sorry for our negligence. References 32, 33 and 26 seem truly not fit the topic described. We have changed the reference 32 and deleted the reference 33, and we have replaced the reference 26 with the new reference 33 throughout the manuscript to keep consistent.
All the modifications are highlighted using yellow background in the revised manuscript.
We tried our best to improve the manuscript and made some changes in the manuscript, however, if the reviewers are not satisfied with that, we could make the further revision. Finally, we are grateful to the reviewer for his (her) critical reading of this manuscript and constructive suggestions. All the questions are important and helpful to improve our manuscript. We hope that the changes incorporated in the revised manuscript could be acceptable.
Best regards
Xiaofeng Xu.
Reviewer 3 Report
referee report
materials-487616
Yi-tong Wang, Yu-guang Zhao, Xiao-feng Xu, Dong Pan, Wen-qiang Jiang and Xue-ying Chong
Simultaneously enhanced strength and ductility of 3 Al-Mg-Si alloys during aging process induced by 4 electro-pulsing treatment
This manuscript describes caharacterization of Al-Mg-Si alloys obtained after electro-pulsing treatment. The authors have applied several characterisation methods including TEM, EBSD and APT to provide deeper information and to support their conclusions.
Furthermore, finite element modelling was carried out. The topic is very interesting, and well suited to be published in Materials.
However, there are some important corrections to be made prior to publication:
(1) Please check all spaces between text and brackets, text and citations, text and numbers.
(2) The English requires some improvement. Do not use abbreviations like what's, isn't etc. within the text.
(3) Many figures are well prepared, but on the TEM figures in Fig. 5, the inserted text is too small and the color should be more contrasting.
(4) For the EBSD data, please provide information on the image quality of the Kikuchi patterns and the indexation quality.
(5) There is no color code for the KAM images (Fig. 4 d-f).
Overall, the present manuscript can be published in Materials after major revision.
Author Response
Dear Editors and Reviewers:
Thank you for your letter and for the reviewer's comments concerning our manuscript entitled "Simultaneously enhanced strength and ductility of Al-Mg-Si alloys during aging process induced by electro-pulsing treatment" submitted to Materials (No: materials-487616). Those comments are all valuable and very helpful for revising and improving our paper, as well as the important guiding significance to our researches. We have studied comments carefully and have made correction which we hope meet with approval. Revised portion are marked in yellow in the paper. The following lists our point-to-point responses to the reviewer's comments and the changes made in the revised manuscript.
Responds to the reviewer's comments:
Reviewer #3:
(1) Please check all spaces between text and brackets, text and citations, text and numbers.
Response: Thanks for your suggestion. We are very sorry for our negligence. We have checked all spaces between text and brackets, text and citations, text and numbers and revised the inappropriate statement. We will be careful not to make such mistakes in the future.
(2) The English requires some improvement. Do not use abbreviations like what's, isn't etc. within the text.
Response: Thanks for your suggestion. We are very sorry for our negligence. We have carefully revised this paper. Some inappropriate interpretations were also corrected. We will be careful not to make such mistakes in the future.
(3) Many figures are well prepared, but on the TEM figures in Fig. 5, the inserted text is too small and the color should be more contrasting.
Response: Thanks for your suggestion. We are very sorry for our negligence. We have enlarged the inserted text and changed the color so that it can be seen clearly.
(4) For the EBSD data, please provide information on the image quality of the Kikuchi patterns and the indexation quality.
Response: Thanks for your suggestion. We are very sorry for our negligence. We have put the band contrast image and MAD image below (Fig. 1 in this paper), and the image quality can be represented by the band contrast image, the indexation quality can be represented by the MAD image. In the process of EBSD acquisition, the indexing percentage of S sample, T6 sample and S+EA sample are 95.48%, 96.25%, 97.35%, respectively, which have been added to the revised manuscript.
Fig. 1. The band contrast image of samples in different states: (a) S, (b) T6, (c) S+EA; and the MAD image of samples in different states: (d) S, (e) T6, (f) S+EA.
(5) There is no color code for the KAM images (Fig. 4 d-f).
Response: Thanks for your suggestion. We are very sorry for our negligence. We have added the color code for the KAM images. We will be careful not to make such mistakes in the future.
All the modifications are highlighted using yellow background in the revised manuscript.
We tried our best to improve the manuscript and made some changes in the manuscript, however, if the reviewers are not satisfied with that, we could make the further revision. Finally, we are grateful to the reviewer for his (her) critical reading of this manuscript and constructive suggestions. All the questions are important and helpful to improve our manuscript. We hope that the changes incorporated in the revised manuscript could be acceptable.
Best regards
Xiaofeng Xu.
Round 2
Reviewer 1 Report
The added reference [27] for formula (3) is not correct. Besides, this formula is not appropriate for the contribution of clusters to YS because strengthening mechanisms by clusters and particles are different. the Orowan mechanism is related to the dislocation bowing between particles, while clusters are cut by dislocations.
Author Response
Dear Editors and Reviewers:
Thank you for your letter and for the reviewer's comments concerning our manuscript entitled "Simultaneously enhanced strength and ductility of Al-Mg-Si alloys during aging process induced by electro-pulsing treatment" submitted to Materials (No: materials-487616). Those comments are all valuable and very helpful for revising and improving our paper, as well as the important guiding significance to our researches. We have studied comments carefully and have made correction which we hope meet with approval. Revised portion are marked in yellow in the paper. The following lists our point-to-point responses to the reviewer's comments and the changes made in the revised manuscript.
Responds to the reviewer's comments:
Reviewer #1:
The added reference [27] for formula (3) is not correct. Besides, this formula is not appropriate for the contribution of clusters to YS because strengthening mechanisms by clusters and particles are different. the Orowan mechanism is related to the dislocation bowing between particles, while clusters are cut by dislocations.
Response: Thanks very much for the reviewer’s comments. We are sorry for this negligence. We have corrected the formula (3) and added a new reference to formula (3), which is appropriate for the contribution of cluster to YS. We will be careful not to make such mistakes in the future.
All the modifications are highlighted using yellow background in the revised manuscript.
We tried our best to improve the manuscript and made some changes in the manuscript, however, if the reviewers are not satisfied with that, we could make the further revision. Finally, we are grateful to the reviewer for his (her) critical reading of this manuscript and constructive suggestions. All the questions are important and helpful to improve our manuscript. We hope that the changes incorporated in the revised manuscript could be acceptable.
Best regards
Xiaofeng Xu.
Reviewer 2 Report
The main point of criticism remains the EBSD measurement. The authors declare, that they made sure “that all the phenomenon measured by EBSD is universal and we choose the most representative one in the manuscript”. At once, they say they “haven’t collected EBSD figures containing more grains”. Please explain, how to ensure that the measurement is representative. The inverse pole figures, which are shown, refer only to the small measurement area and cannot be applied to the whole sample.
After revision, the grain size discussion was left out due to the lack of EBSD measurements at lower magnification. Grain size measurements can also be done using optical micrographs. Analysis via optical microscopy is not time consuming and would give an overview over the microstructure (without orientations, but with regard to grain size).
The authors interpret the black spots as subgrain boundaries. It should be carefully checked whether this is not the influence of the measurement (measurement noise). Highest amount of misorientation angle is below 0.5. A noise reduction should be performed on the original EBSD dataset, like it is recommended in Hildyard et al., Electron Backscatter Diffraction (EBSD) Analysis of Bassanite Transformation Textures and Crystal Structure Produced from Experimentally Deformed and Dehydrated Gypsum, Journal of Petrology, Volume 52, Issue 5, May 2011, Pages 839–856, https://doi.org/10.1093/petrology/egr004 or in https://www-user.tu-chemnitz.de/~rahi/mtexWorkshop16/presentations/denoising.pdf.
The EBSD measurement is the core of the authors’ interpretation and should be carried out and analyzed very carefully.
Author Response
Dear Editors and Reviewers:
Thank you for your letter and for the reviewer's comments concerning our manuscript entitled "Simultaneously enhanced strength and ductility of Al-Mg-Si alloys during aging process induced by electro-pulsing treatment" submitted to Materials (No: materials-487616). Those comments are all valuable and very helpful for revising and improving our paper, as well as the important guiding significance to our researches. We have studied comments carefully and have made correction which we hope meet with approval. Revised portion are marked in yellow in the paper. The following lists our point-to-point responses to the reviewer's comments and the changes made in the revised manuscript.
Responds to the reviewer's comments:
Reviewer #2:
The main point of criticism remains the EBSD measurement. The authors declare, that they made sure “that all the phenomenon measured by EBSD is universal and we choose the most representative one in the manuscript”. At once, they say they “haven’t collected EBSD figures containing more grains”. Please explain, how to ensure that the measurement is representative. The inverse pole figures, which are shown, refer only to the small measurement area and cannot be applied to the whole sample.
After revision, the grain size discussion was left out due to the lack of EBSD measurements at lower magnification. Grain size measurements can also be done using optical micrographs. Analysis via optical microscopy is not time consuming and would give an overview over the microstructure (without orientations, but with regard to grain size).
The authors interpret the black spots as subgrain boundaries. It should be carefully checked whether this is not the influence of the measurement (measurement noise). Highest amount of misorientation angle is below 0.5. A noise reduction should be performed on the original EBSD dataset, like it is recommended in Hildyard et al., Electron Backscatter Diffraction (EBSD) Analysis of Bassanite Transformation Textures and Crystal Structure Produced from Experimentally Deformed and Dehydrated Gypsum, Journal of Petrology, Volume 52, Issue 5, May 2011, Pages 839–856, https://doi.org/10.1093/petrology/egr004 or in https://www-user.tu-chemnitz.de/~rahi/mtexWorkshop16/presentations/denoising.pdf.
The EBSD measurement is the core of the authors’ interpretation and should be carried out and analyzed very carefully.
Response: Thanks very much for the reviewer’s comments. Firstly, the term that we declared before “that all the phenomenon measured by EBSD is universal and we choose the most representative one in the manuscript” means that in the measurement of EBSD, we tested several areas to ensure the repeatability of the experiment, and because of the space limitation, we chose the most representative one in the manuscript. And the term that “haven’t collected EBSD figures containing more grains” means that each EBSD data is of the same magnification times as the data in the manuscript, so in each EBSD data, we haven’t collected EBSD figures containing more grains at a lower magnification.
Secondly, as for the black spots, we misunderstood the question the reviewer gave in the last comments that “grey lines, which should represent grain boundaries less than 2°, are not visible (g), (h), (i).” We thought that the gray lines couldn’t be seen because of the color, so we changed the color from gray line to black lines. In our EBSD data of the manuscript, the noise reduction has been truly performed on the original EBSD dataset from the noise reduction function of the CHANNEL 5 software, as we changed the color of sub grain boundaries back to gray lines, the gray lines in the IPFs can be corresponded to the black lines in the KAM.
Thirdly, as for the grain size, we followed the reviewer’s suggestion and have given the optical microscopy in the Fig. 4 of the revised manuscript.
Finally, as for the inverse pole figures, we followed the reviewer’s suggestion, they refer only to the small measurement area and cannot be applied to the whole sample, so we have deleted the inverse pole figures from Fig. 4.
What’s more, because the addition of optical microscopy, considering of the space limitation, and we can clearly see the differences of low misorientation in the KAM images, we have deleted the histogram of the low misorientation. Thanks again for the reviewer’s constructive suggestions.
All the modifications are highlighted using yellow background in the revised manuscript.
We tried our best to improve the manuscript and made some changes in the manuscript, however, if the reviewers are not satisfied with that, we could make the further revision. Finally, we are grateful to the reviewer for his (her) critical reading of this manuscript and constructive suggestions. All the questions are important and helpful to improve our manuscript. We hope that the changes incorporated in the revised manuscript could be acceptable.
Best regards
Xiaofeng Xu.
Reviewer 3 Report
referee report
materials-487616.R1
Yi-tong Wang, Yu-guang Zhao, Xiao-feng Xu, Dong Pan, Wen-qiang Jiang and Xue-ying Chong
Simultaneously enhanced strength and ductility of 3 Al-Mg-Si alloys during aging process induced by 4 electro-pulsing treatment
Points from the previous report:
(1) Please check all spaces between text and brackets, text and citations, text and numbers, quantities and units.
There are still several problems left in the revised version. E.g.,
-- p.2, line 78: space. Also, the sign used for ° is not the correct ASCII one.
-- p.3, line 85: the degree sign here looks very funny.
-- p.3, line 85: here it is correct!
-- p.4, line 118: What is this for an unit? If it is a multiplication, please use "x".
-- p.5, line 131: The subscripts in the chemical formulae do not look normal -- these are not subscripts, but just small letters.
-- p.5, line 147: What is this for a sign -- and underlined degree?!
-- p. 9, line 222: There is a sign for diameter in the extended ASCII code, but not this one.
-- p. 12, line 317: Burgers is a name -- captial B!
-- p. 12, line 317: Check the spaces in the formula.
(2) The English requires some improvement. Do not use abbreviations like what's, isn't etc. within the text.
The changes made have improved the quality of the manuscript. Still, there are several places which require attention, e.g.,
--"tangles" -- this expression does not have a plural.
-- p.2, line 79: referred to as...
-- Table 3: each cluster
-- some of the abbreviations are still present, e.g., p. 9, line 231.
-- Check the use of defined and undefined articles throughout the manuscript.
-- Please let the manuscript be checked from a native speaker.
(3) Many figures are well prepared, but on the TEM figures in Fig. 5, the inserted text is too small and the color should be more contrasting.
-- Fig. 5 was treated well.
-- Fig. 3 is still too small concerning the text.
-- Fig. 7 has not a good resolution, it looks blurred. Please plot the data using a real drawing program, and not simply the
data from the analysis software. All lettering is too small for publication.
-- Fig. 8 a,b,i: The axis indication is too small to be read. I need 400% magnification to see it! And j is impossible!
-- Similar comments apply to parts of Fig. 9.
(4) For the EBSD data, please provide information on the image quality of the Kikuchi patterns and the indexation quality.
-- no information given. The EBSD software provides this. So, how is the image quality (IQ) and the confidence index (CI)?
(5) There is no color code for the KAM images (Fig. 4 d-f).
done.
And, there is a new point:
(6) The equations included in the paper are difficult to be read. What kind of math editor was used for this purpose?
Please check all spaces, subscripts,...
Equations 3 and 4: What is the subscript for? It does not appear in the main text... Check the spaces between G and b.
Eq. 4.: What purpose has the dash above the "/"?
Overall, the present revision is insufficient. Please take time to produce a proper revision, hectic does not help. The points mentioned here
are NOT complete, there are still many small points which require attention.
Author Response
Dear Editors and Reviewers:
Thank you for your letter and for the reviewer's comments concerning our manuscript entitled "Simultaneously enhanced strength and ductility of Al-Mg-Si alloys during aging process induced by electro-pulsing treatment" submitted to Materials (No: materials-487616). Those comments are all valuable and very helpful for revising and improving our paper, as well as the important guiding significance to our researches. We have studied comments carefully and have made correction which we hope meet with approval. Revised portion are marked in yellow in the paper. The following lists our point-to-point responses to the reviewer's comments and the changes made in the revised manuscript.
Responds to the reviewer's comments:
Reviewer #3:
Points from the previous report:
(1) Please check all spaces between text and brackets, text and citations, text and numbers, quantities and units.
There are still several problems left in the revised version. E.g.,
-- p.2, line 78: space. Also, the sign used for ° is not the correct ASCII one.
-- p.3, line 85: the degree sign here looks very funny.
-- p.3, line 85: here it is correct!
-- p.4, line 118: What is this for an unit? If it is a multiplication, please use "x".
-- p.5, line 131: The subscripts in the chemical formulae do not look normal -- these are not subscripts, but just small letters.
-- p.5, line 147: What is this for a sign -- and underlined degree?!
-- p. 9, line 222: There is a sign for diameter in the extended ASCII code, but not this one.
-- p. 12, line 317: Burgers is a name -- captial B!
-- p. 12, line 317: Check the spaces in the formula.
Response: Thanks for your comments. We are sorry for our mistake. We have corrected the term to the revised manuscript. We will be careful not to make such mistakes in the future. As for the question that “p.5, line 131: The subscripts in the chemical formulae do not look normal -- these are not subscripts, but just small letters.”, after carefully examining the subscripts in chemical formulae, we found that these are truly subscripts, not small letters, which can be seen in the Fig. 1 (in this paper) below.
Fig. 1. The format of the subscripts in the chemical formulae
(2) The English requires some improvement. Do not use abbreviations like what's, isn't etc. within the text.
The changes made have improved the quality of the manuscript. Still, there are several places which require attention, e.g.,
--"tangles" -- this expression does not have a plural.
-- p.2, line 79: referred to as...
-- Table 3: each cluster
-- some of the abbreviations are still present, e.g., p. 9, line 231.
-- Check the use of defined and undefined articles throughout the manuscript.
-- Please let the manuscript be checked from a native speaker.
Response: We are very sorry for our negligence. We have corrected the corresponding term to the revised manuscript. The question is important and helpful to improve our manuscript. Thanks for the reviewer’s reminding.
(3) Many figures are well prepared, but on the TEM figures in Fig. 5, the inserted text is too small and the color should be more contrasting.
-- Fig. 5 was treated well.
-- Fig. 3 is still too small concerning the text.
-- Fig. 7 has not a good resolution, it looks blurred. Please plot the data using a real drawing program, and not simply the
data from the analysis software. All lettering is too small for publication.
-- Fig. 8 a,b,i: The axis indication is too small to be read. I need 400% magnification to see it! And j is impossible!
-- Similar comments apply to parts of Fig. 9.
Response: Thanks for the reviewer’s suggestion. We have enlarged the inserted text and the corresponding axis indication in the revised manuscript to satisfy the requirement of publication.
(4) For the EBSD data, please provide information on the image quality of the Kikuchi patterns and the indexation quality.
-- no information given. The EBSD software provides this. So, how is the image quality (IQ) and the confidence index (CI)?
Response: Thanks for your comments. After carefully consult about the reference about EBSD data (Koblischka-Veneva, A; Koblischka, M.R.; Mucklich, F. Advanced microstructural analysis of ferrite materials by means of electron backscatter diffraction (EBSD). Journal of Magnetism and Magnetic Materials. 2010, 322, 1178-1181. https://doi.org/10.1016/j.jmmm.2009.06.073.), we found that the image quality (IQ) and the confidence index (CI) were obtained by the EDAX-TSL OIM analysis unit. But our EBSD data was obtained by OXFORD CHANNEL 5 software. After consulting with the engineer of OXFORD, the image quality (IQ) and the confidence index (CI) cannot be obtained from the CHANNEL 5 software. But in our (the authors’ and the engineer’s) opinion, the image quality (IQ) can be represented by the indexing percentage, which means the proportion that can be calibrated during the data collection process, the indexing percentage of S sample, T6 sample and S+EA sample are 95.48%, 96.25%, 97.35%, respectively. The confidence index (CI) can be represented by the mean angular deviation (MAD), which was also obtained during the data collection process. The MAD of S, T6 and S+EA sample are 0.62, 0.58, 0.56, respectively.
(5) There is no color code for the KAM images (Fig. 4 d-f).
done.
And, there is a new point:
(6) The equations included in the paper are difficult to be read. What kind of math editor was used for this purpose?
Please check all spaces, subscripts,...
Equations 3 and 4: What is the subscript for? It does not appear in the main text... Check the spaces between G and b.
Eq. 4.: What purpose has the dash above the "/"?
Response: Thanks for your comments. We are sorry for our mistake. We have corrected the corresponding term to the revised manuscript. We will be careful not to make such mistakes in the future.
All the modifications are highlighted using yellow background in the revised manuscript.
We tried our best to improve the manuscript and made some changes in the manuscript, however, if the reviewers are not satisfied with that, we could make the further revision. Finally, we are grateful to the reviewer for his (her) critical reading of this manuscript and constructive suggestions. All the questions are important and helpful to improve our manuscript. We hope that the changes incorporated in the revised manuscript could be acceptable.
Best regards
Xiaofeng Xu.
Round 3
Reviewer 2 Report
The authors have endeavored to improve the paper. This can be regarded as successful at some points. However, with regard to the EBSD results unfortunately I cannot agree to publish the paper in the current state. In my first review, I used a printed version, so I was not able to see the grey lines, which were attributed to sub-grain boundaries by the authors. After changing the color it was better for me to see, what the authors mean by sub-grain boundaries. However, I do not agree with the authors, that this dots and small lines can be explained as sub-grain boundaries. The optical micrographs help to get an overview of the microstructure. And as can be seen, there are some inclusions or contaminations. Maybe they have an influence on the EBSD results. Furthermore, the grey lines seem to follow a straight way (for example from scratches). And even if the misorientation angle distribution was omitted, the angles were lower than 0.5° and it is most likely, that this effect results from measurement noise. As mentioned before, the EBSD results are the core of the authors’ discussion and should be more carefully elaborated.
Reviewer 3 Report
referee report
materials-487616.R2
Yi-tong Wang, Yu-guang Zhao, Xiao-feng Xu, Dong Pan, Wen-qiang Jiang and Xue-ying Chong
Simultaneously enhanced strength and ductility of 3 Al-Mg-Si alloys during aging process induced by 4 electro-pulsing treatment
The present revision of the manuscript is well done; especially the changes to the figures improve the quality of the
manuscript. Although some minor issues to the English still remain, the manuscript is now suitable for publication.